# Wide-Horizon Thinking and Simulation-Based Evaluation for Real-World LLM Planning with Multifaceted Constraints

**Dongjie Yang**[1], **Chengqiang Lu**[2], **Qimeng Wang**[2],
**Xinbei Ma**[1], **Yan Gao**[2], **Yao Hu**[2], **Hai Zhao**[1,*]
[1]Shanghai Jiao Tong University, [2] Xiaohongshu Inc.
[1]{djyang.tony@,sjtumaxb@,zhaohai@cs.}sjtu.edu.cn,
[2]{lusuo,haoli9,yadun,xiahou}@xiaohongshu.com

## Abstract

Unlike reasoning, which often entails a deep sequence of deductive steps, complex real-world planning is characterized by the need to synthesize a broad spectrum of parallel and potentially conflicting information and constraints. For example, in travel planning scenarios, it requires the integration of diverse real-world information and user preferences. While LLMs show promise, existing methods with long-horizon thinking struggle with handling multifaceted constraints, leading to suboptimal solutions. Motivated by the challenges of real-world travel planning, this paper introduces the Multiple Aspects of Planning (MAoP), empowering LLMs with "wide-horizon thinking" to solve planning problems with multifaceted constraints. Instead of direct planning, MAoP leverages the strategist to conduct pre-planning from various aspects and provide the planning blueprint for planners, enabling strong inference-time scalability by scaling aspects to consider various constraints. In addition, existing benchmarks for multi-constraint planning are flawed because they assess constraints in isolation, ignoring causal dependencies within the constraints, e.g, travel planning, where past activities dictate future itinerary. To address this, we propose Travel-Sim, an agent-based benchmark assessing plans via real-world simulation, thereby inherently resolving these causal dependencies. This paper advances LLM capabilities in complex planning and offers novel insights for evaluating sophisticated scenarios through simulation.

## 1 Introduction

Large Language Models (LLMs) [1, 2, 3, 4] have shown significant promise in planning by generating action sequences to achieve specified goals. However, transitioning from controlled environments to the complexities of real-world planning presents a formidable challenge [5, 6, 4], defined by the need to manage a multitude of diverse and simultaneous constraints. Prevailing methods [7, 8] often rely on task decomposition, a linear and sequential methodology that breaks a complex problem into simpler sub-tasks. This approach is effective in domains with limited constraints, such as in GUI automation [9, 10], but its feasibility diminishes sharply in real-world scenarios where constraints are deeply interconnected. The failure of this step-by-step strategy highlights a fundamental mismatch between the problem-solving approach and the non-linear nature of the challenge itself.

The cognitive model underpinning this sequential approach is analogous to that of logical and mathematical reasoning [11, 12, 13]. These domains epitomize long-horizon thinking: a deep, step-by-step deductive process that the entire chain of reasoning funnels toward a limited set of deterministic

---

[*] Corresponding author.

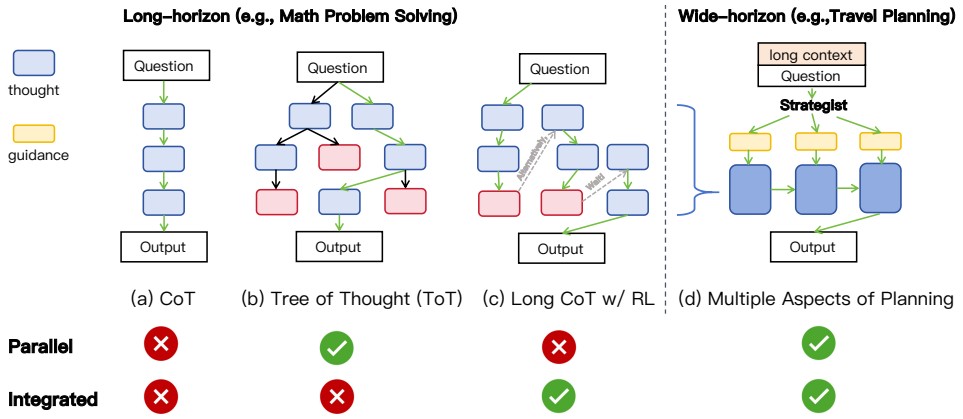

Figure 1: The comparison between long-horizon and wide-horizon thinking reveals distinct cognitive approaches. While long-horizon thinking involves deep exploration of a single reasoning trajectory, wide-horizon thinking incorporates heterogeneous information and constraints in long contexts by considering various aspects. It necessitates **parallel** consideration of multiple dimensions, which are subsequently **integrated** to generate comprehensive outputs.

outcomes. Real-world planning [6, 14, 15], however, demands a fundamentally different cognitive paradigm. It requires an LLM to move beyond linear deduction to instead simultaneously integrate multifaceted constraints and heterogeneous information, necessitating wide-horizon thinking from various aspects rather than long-horizon thinking. Success hinges less on the deductive soundness of a sequence and more on the holistic feasibility of a plan that concurrently satisfies a wide array of interconnected variables.

In this paper, we explore the limitations and potential of LLMs in real-world planning through the lens of the travel planning scenario. We conduct preliminary experiments to investigate the LLM's zero-shot capability of solving the travel planning problem. Alongside long-horizon baselines, we implement a simple wide-horizon approach based on aspect-aware question decomposition. Our results indicate that LLMs struggle to address real-world planning problems using long-horizon thinking, performing notably better with even a naive wide-horizon method.

Our preliminary experiments also reveal that this naive wide-horizon approach, which relies on decomposing the request into various aspects, has its own significant limitations: 1) lack of inter-aspect associations; 2) dependence on well-crafted artifactual guidance; 3) poor inference-time scalability. To address these limitations, we propose **"Multiple Aspects of Planning"** (**MAoP**). In MAoP, instead of directly planning, we introduce a strategist to conduct the pre-planning using a two-stage strategy. The strategist first analyzes the context and **decomposes** the planning request into various aspects to be considered. For each aspect, the strategist provides short guidance for the planner to further analyze subsequently. In the second **routing** stage, the strategist integrates the independent aspects into a coherent planning blueprint, leading to better inference-time scaling performance with more aspects considered. In the actual planning of MAoP, the planner concentrates on guidance of a single aspect per dialogue turn by following the pre-planning blueprint, progressively constructing a comprehensive planning process via multiple turns of dialogue.

While MAoP improves real-world planning, the evaluation of multi-constraint planning remains a major hurdle. The challenges (e.g, from travel planning) are twofold: 1) The subjective nature of travel planning means there is no universal optimal solution, as users weigh constraints like cost and convenience differently, making objective evaluation elusive. 2) The causal dependency within a journey means a single failure can dynamically violate a cascade of subsequent, interconnected constraints, rendering simple, static evaluations inadequate. Previous studies [5, 6, 16] introduce constraint pass rates as metrics, which only reflect the partial feasibility of the plan and the satisfaction of coarse-grained user requirements. These metrics ignore the real-world influence and causal consistency in real travel scenarios, poorly reflecting actual feasibility.

As the proof is in the pudding, we propose a novel agent-based evaluation framework to simulate a trip based on the real-world environment. At its core, an LLM-powered traveler agent executes a plan within an event-driven sandbox. By leveraging live traffic data from maps and qualitative insights

from travel blogs, we enable the simulation to organically capture the dynamics and unforeseen events of real-world scenarios. To better emulate real-world individuals, the traveler agent is meticulously designed with diverse personas and a dynamic stamina engine that simulates physical exertion. Beyond the "static" metrics (constraint pass rates), we introduce "dynamic" metrics that the traveler evaluates the plan based on their experience, offering multi-granularity feedback to assess the experience at multiple levels.

Our contributions can be concluded as follows:

- We propose MAoP to enhance wide-horizon thinking capabilities for solving real-world planning with multifaceted constraints.

- We propose a simulation-based evaluation framework to evaluate the feasibility and appeal of the travel plan, offering novel insights for assessing complex scenarios.

## 2 Preliminaries

### 2.1 Long-Horizon vs. Wide-Horizon

As shown in Figure 1, previous studies [1, 17, 7] have primarily centered on developing CoT methods and their variants to solve planning and reasoning problems requiring deep exploration along a single trajectory. Deepseek-R1 [18] further enhances the long-horizon reasoning ability by leveraging Reinforcement Learning. In contrast, real-world planning such as travel planning necessitates LLMs to: 1) extract pertinent information from long contexts (e.g., tour guides, spatial information, traveler information, etc); 2) conduct deliberate thinking over multifaceted constraints (e.g., real-world constraints and preferences). Travel planning problem does not require the model to reason deeply, but necessitates considering multiple aspects simultaneously in a wide-horizon view. In this paper, we investigate the potential of conducting wide-horizon thinking compared to long-horizon thinking in real-world planning.

### 2.2 Wide-Horizon Thinking with Aspect-Aware Guidance

We conducted a preliminary experiment to compare the efficacy of long-horizon versus wide-horizon thinking on real-world planning with multifaceted constraints. The input provided to the model comprises two components: a rich context and a structured guidance prompt. The context includes real-world information such as traveler profiles, travel blogs, and spatial data (see Appendix C for processing details). The guidance is a carefully crafted instruction that outlines the key aspects to consider when generating a travel plan. To establish our long-horizon thinking baselines, we use this guidance to prompt two methods: zero-shot CoT and the Plan-and-Solve framework [1], which relies on task decomposition. Our approach to naive wide-horizon thinking is centered on aspect-based decomposition. Critically, unlike prior works [20, 2] that decompose a question into a sequence of simpler sub-tasks, our method

Table 1: The comparison between long-horizon thinking and wide-horizon thinking in travel planning by evaluating the Feasibility (FEA) and Personalization (PER) scores in Travel-Sim.

| Method | FEA | PER |
|---|---|---|
| Qwen 2.5-32B [19] | | |
|     w/ Artificial Guidance + CoT | 23.3 | 36.2 |
|     w/ Artificial Guidance + Plan&Solve | 25.0 | 39.7 |
|     w/ Artificial Guidance + Wide | **31.9** | **44.1** |
|     w/ Self-Gen. Guidance + Wide | 29.4 | 42.0 |
| DeepSeek-R1 [18] | | |
|     w/ Artificial Guidance + CoT | 52.6 | 62.5 |
|     w/ Artificial Guidance + Plan&Solve | 57.4 | 64.2 |
|     w/ Artificial Guidance + Wide | **58.9** | **68.0** |
|     w/ Self-Gen. Guidance + Wide | 55.2 | 64.1 |

breaks the guidance into multiple aspects designed to be considered concurrently. The LLM then independently analyzes each aspect and finally synthesizes the insights from these parallel analyses to generate an integrated output. In Table 1, we find that naive wide-horizon thinking with aspect-aware guidance significantly improves the travel plan quality with better feasibility and personalization scores. Although the wide-horizon thinking demonstrates significant potential, well-designed artificial guidance can further enhance the performance over self-generated guidance. More details can be checked in Section 5 and Appendix D.

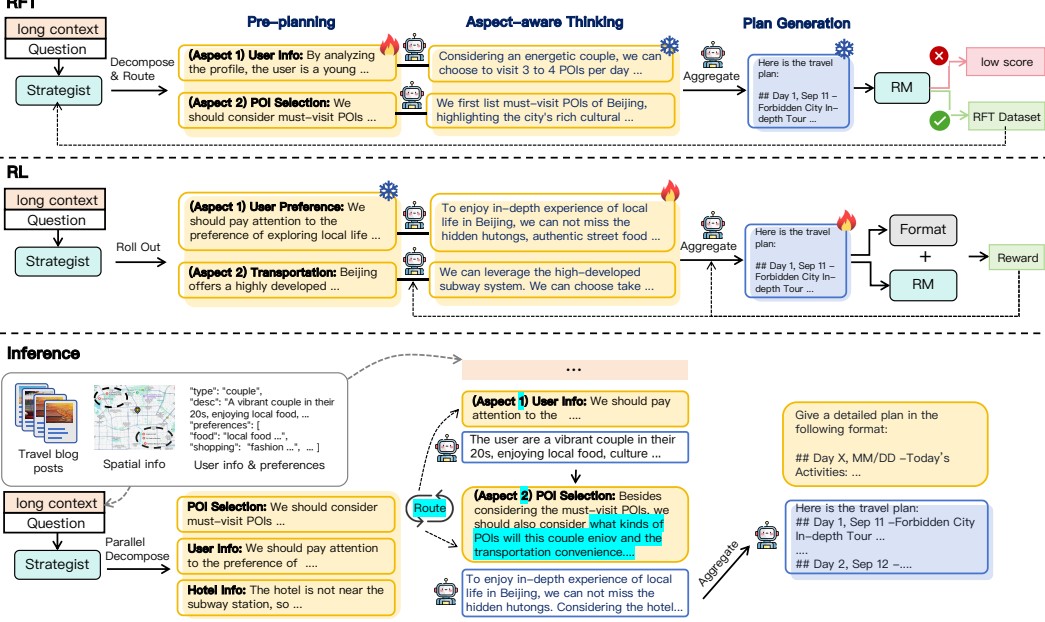

Figure 2: The overview of the MAoP training and inference process.

## 3 Wide-Horizon Thinking on Real-world Planning

Based on our preliminary experiments, we find that while the naive wide-horizon thinking approach shows promise, it still exhibits several key deficiencies: 1) Although aspect-aware thinking is concurrent, its independence inherently limits the capture of inter-aspect associations; 2) the artificial guidance demonstrates superiority, yet its deliberate design remains challenging to implement; 3) as the number of aspects increases, the performance can not be scaled with more inference-time compute. Therefore, we propose the Multiple Aspects of Planning (MAoP), a method that leverages a strategist to conduct the pre-planning to address these deficiencies in naive wide-horizon thinking.

### 3.1 MAoP Training

The training of MAoP consists of three stages: reward model training, rejection sampling finetuning for the strategist [2], and RL training for the planner. As shown in Figure 2, we mainly focus on the last two stages and provide more details in Appendix E.

**Rejection Sampling Finetuning (RFT) for Strategist**    In MAoP, as shown in Figure 2, the strategist mainly does two things in pre-planning: 1) **decomposing** the original instruction and generating the aspect-aware guidance; 2) **routing** the planning trajectories to capture inter-aspect associations. These core operations solve the mentioned deficiencies in naive wide-horizon thinking. To train such a strategist, we implement RFT while keeping the planner frozen. For each request, we prompt the strategist to first conduct pre-planning for $N$ times, and subsequently the planner to generate final plans. The rejection strategy is to reject the trajectory where all $N$ plans, as defined in Eq. 1, fall below a predefined threshold.

**RL Training for Planner**    Following an initial cold-start RFT, we conduct RL training using GRPO [11] on the planner to further improve aspect-aware thinking ability. To mitigate reward hacking, we design a multi-dimensional reward function. It is primarily guided by the PER score (a composite of five criteria, see Sec 5.1 and Appendix E) and also includes an auxiliary reward for proper plan formatting. The overall function is defined as follows:

---

[2]We do not further implement RL on the strategist. The RL pipeline has to go through a frozen planner to get rewards, making it hard to optimize.

$$R_{overall} = \begin{cases} 2(R_{PER} - 0.5), & \text{if the format is correct} \\ 2(R_{PER} - 0.5) - 1, & \text{if the format is incorrect} \end{cases}, \quad \text{where } R_{PER} \in [0, 1]. \quad (1)$$

## 3.2 MAoP Inference

### 3.2.1 Pre-Planning

**Decomposition** As shown in Figure 2, the strategist accepts the long context and question as the input and decomposes the planning request into various aspects. For each aspect, the strategist generates concise guidance to instruct the planner how to conduct a thorough analysis. By parallel sampling the strategist multiple times, we can derive a large amount of aspect-guidance pairs.

**Routing** Different from the naive wide-horizon thinking that treats the aspects independently, the strategist additionally selects and aggregates the aspects to route the best planning blueprint. From the experiments in Appendix D.2, we find that if we directly increase the number of aspects by sampling the strategist more times, the planner only benefits from considering 3 ~5 aspects without further inference-time scaling capability. It is because more aspects introduce more information but also more noise. To address this, the strategist aggregates the number of aspects into a smaller number and constructs a coherent planning blueprint. As shown in Figure 2, subsequent aspect guidance is determined through the influence of preceding multiple aspects. The routing process shifts the burden of considering numerous aspects from the planner to the strategist, thus enabling the **scalability of considering more aspects to further improve the wide-horizon thinking**.

### 3.2.2 Aspect-Aware Thinking

Based on the planning blueprint constructed by the strategist, the planner sequentially conducts thinking over the aspects in a coherent multi-turn dialogue. With aspect-specific guidance, the planner can conduct a more focused and in-depth analysis over the long context from this aspect. After multiple turns of profound analysis, the planner produces the final plan based on the previous wide-horizon thoughts in the last turn of the conversation.

## 3.3 MAoP Distillation for One-Step Wide-Horizon Thinking

Implementing MAoP is complex, involving two models and a multi-turn planning process. We accelerate and simplify this process by distillation. To create high-quality training data for distillation, we employ a powerful teacher model, composed of a strategist and a planner, to generate MAoP samples. From these generated planning trajectories, we extract the strategist's aspect-specific guidance and then compress the entire aspect-aware thinking and the final aggregation into a single, consolidated output. By finetuning on this distilled data, the model learns to execute complex MAoP planning in a single inference step. This capability is what we term one-step wide-horizon thinking.

# 4 Causal Evaluation via Agent-based Simulation

As shown in Table 2, previous benchmarks typically rely on static, rule-based metrics, such as pass rates for individual constraints. However, this approach overlooks a fundamental truth: travel is a dynamic, causal process, not a static checklist. Each event, from a delayed train to physical exhaustion on the first day, directly impacts the feasibility and enjoyment of the rest of the journey. By neglecting these critical causal dependencies, existing benchmarks fail to adequately evaluate a plan's real-world viability and its capacity to meet a traveler's evolving personal needs.

We introduce Travel-Sim, a novel benchmark framework utilizing agent-based simulation. In this framework, a traveler agent, embodied by the advanced Gemini 2.5-Pro-Exp-0325 [21], simulates a journey according to a given travel plan. Throughout the simulation, the agent provides continuous, experience-driven feedback. This dynamic, simulation-based approach inherently resolves the causal dependency issues of static benchmarks while capturing personalized, multi-granularity evaluations from an authentic traveler's perspective.

Table 2: Comparison of evaluation metrics for travel planning between different benchmarks.

| Method | Rule-based | LLM-Judge | Multi-Granularity | Causality |
|---|---|---|---|---|
| TravelPlanner [5] | ✓ | | ✓ | |
| UnsatChristmas [22] | ✓ | | ✓ | |
| TravelAgent [6] | | ✓ | | |
| ITINERA [23] | ✓ | ✓ | ✓ | |
| ChinaTravel [16] | ✓ | | ✓ | |
| Travel-Sim **(ours)** | ✓ | ✓ | ✓ | ✓ |

## 4.1 Travel Experience Simulation

To simulate a realistic travel experience, we build a sandbox to provide travelers with any real-world information they need, such as time, location, transportation, and even sightseeing experiences from blog posts. Similar to role-playing, we set up detailed profiles for traveler agents, including group sizes, character types, ages, genders, budgets, and preferences. Moreover, we design a stamina engine for the traveler. Stamina is closely related to character types (e.g., young people vs. elderly, family w/ baby vs. family w/o baby) and has a significant impact on the travel experience. The travelers of different types have their own rules for stamina exertion when encountering different events.

**Event-Driven State Transition** The traveler follows the plan $p$ to travel in the sandbox. We define the traveler's state as $c_n = \{t, l, s, o, e\}$ at step $n$, where $t, l, s, o, e$ represent the current *time*, *location*, *stamina*, *outlay*, and the ongoing *event*. The traveler employs a policy $\pi(a_n|c_1, c_2, \cdots, c_{n-1}, p)$, where action $a_n \in \mathcal{A}$ and action space $\mathcal{A}$ covers basic actions such as *transiting*, *resting*, *dining*, and *sightseeing*. Each action the traveler takes leads to a new event, which transitions the previous state to the new one. Similar to ReAct [7], the traveler agent first thinks over the current situation from the traveler's perspective and then makes a decision on the next action.

**Real-world Information Integration** When coming to a new city, the traveler usually starts the travel at the train station or airport. The traveler can choose an action; in most cases, the first step is to proceed to the hotel. We utilize a map API to provide various modes of transportation as references. The traveler comprehensively considers stamina (being tired from the long flight), schedule (next event in the plan), and budget to make a decision. For example, if the traveler opts to take the metro to the hotel, the environment updates the traveler's state with arrival time, new location (hotel), new stamina, etc. Among the events, the simulation of activities like sightseeing or shopping, due to the various experiences and interactions, presents a more significant challenge. Although the traveler agent can not physically visit the POI, we utilize a special event agent to generate how the traveler would do the sightseeing by referring to the real experiences in the travel blog posts.

## 4.2 Multi-granularity Evaluation by Traveler

**Evaluation Process** To capture detailed feedback, we implement a multi-granularity evaluation mechanism. Travelers assess their experience across five core dimensions: experience (ex), interest (it), arrangement (ar), stamina (st), and cost (co) (detailed in Appendix F.4). These evaluations are collected at three distinct levels of granularity: per-POI (after each POI visit), per-day (at the conclusion of each day), and per-trip (upon completion of the entire journey).

**Dataset Construction** We construct a variety of traveler profiles, including 16 distinct types of travelers. Each differs in terms of group size, age, gender, stamina level, and preferences. We carefully select 7 Chinese cities that are ideal for tourism as destinations. We have 112 different distinct {traveler, destination, duration} combinations. See more details in Appendix F.3.

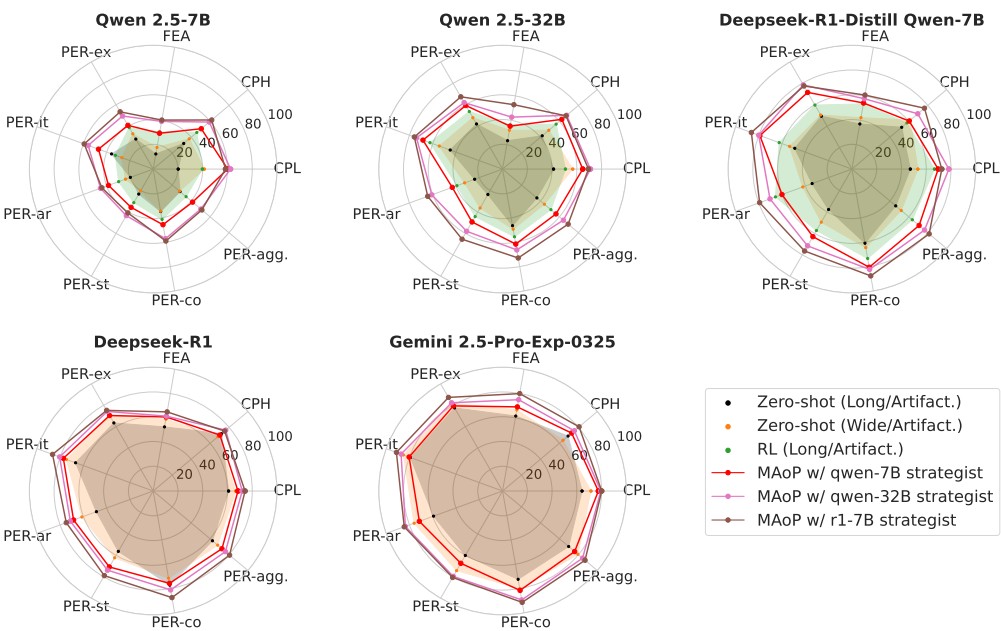

Figure 3: The comprehensive comparison results between MAoP and the baseline methods.

## 5 Experiment

### 5.1 Evaluation Setup

**Methods**  For the baseline, we reuse the setup of preliminary experiments, including zero-shot long-horizon thinking and naive wide-horizon thinking. We add an additional baseline that we implement RL to train the models with *Long/Artifact.* setting using the same training dataset as the MAoP. This additional baseline is to compare the finetuned performance between the long-horizon thinking and the MAoP method. For the MAoP, we train the Qwen 2.5-7B [19], Qwen 2.5-32B [19], and Deepseek-R1-Distill Qwen-7B [18] as the strategists and the planners, respectively. We also include Deepseek-R1 [18] and Gemini-2.5-Pro-Exp-03-25 [21] as additional zero-shot planners. During the MAoP inference, the strategists only route no more than 8 aspects into the planning blueprint for fair comparison.

To simplify MAoP for one-step wide-horizon thinking, we distill the MAoP synthetic data to train Qwen 2.5-3B [19], Llama 3.2-3B [24], and Deepseek-R1-Distill Qwen-7B. The teachers in the distillation are the Deepseek-R1-Distill Qwen-7B as the strategist and Gemini 2.5-Pro-Exp-0325 as the planner. More training details can be checked in Appendix E.

**Metrics**  We evaluate the MAoP on our Travel-Sim to showcase how MAoP better deals with the travel planning problem via wide-horizon thinking. We introduce four metrics (with details in Appendix G): comprehensiveness (CPH), completeness (CPL), feasibility (FEA), and personalization (PER), where the last two metrics have been used in the preliminary experiment. The first two are rule-based metrics. Comprehensiveness (CPH) evaluates how much relevant information is effectively integrated from the long context into the final plan. Completeness (CPL) evaluates if the travel plan strictly follows the formatting instructions and basic constraints according to the artifactual criteria.

The last two metrics, feasibility (FEA) and personalization (PER), are based on the travel simulation. To evaluate the FEA, also named Travel Plan Similarity Score (TPSS), we develop an algorithm that calculates the similarity between the trajectory from the travel plan and the trajectory from the simulated travel, as shown in Figure 5 and Algorithm 3. This metric measures the discrepancy between what is planned and the actual execution in the simulation. If the planned trajectory is similar to the simulated one, it means the plan is more feasible. The PER is associated with the feedback of the traveler agent after the simulated travel, indicating if the plan is personalized and suitable for this traveler. The PER is a comprehensive metric, including the evaluation from multiple dimensions and

granularities: 1) Multi-dimension: The traveler evaluates the travel experience from five perspectives, i.e., experience (ex), interest (it), arrangement (ar), stamina (st), and cost (co). 2) Multi-granularity: the traveler conducts evaluation from three levels (feedback after visiting every POI, finishing the whole day, and finishing the whole journey). We aggregate the scores from various dimensions and granularities of travel experience, according to Equation 3, to calculate a final PER score.

Table 3: Comparison between MAoP distilled models and MAoP combinations.

| Model | CPH | CPL | FEA | PER | | | | | |
| --- | --- | --- | --- | --- | --- | --- | --- | --- | --- |
| | | | | ex | it | ar | st | co | agg. |
| **MAoP** | | | | | | | | | |
| Qwen 2.5-7B (s.) + Qwen 2.5-32B (p.) | 64.8 | 62.5 | 35.2 | 59.3 | 68.4 | 43.2 | 49.3 | 61.5 | 56.3 |
| R1-Distill 7B (s.) + R1-Distill 7B (p.) | 72.6 | 76.5 | 60.7 | 77.5 | 86.4 | 79.3 | **76.4** | 87.6 | 81.4 |
| **MAoP Distillation** | | | | | | | | | |
| Llama 3.2-3B (Distill) | 61.3 | 59.2 | 52.9 | 62.0 | 65.2 | 63.1 | 62.5 | 70.5 | 65.7 |
| Qwen 2.5-3B (Distill) | 64.2 | 65.8 | 53.1 | 64.2 | 65.9 | 63.4 | 61.4 | 73.5 | 66.9 |
| R1-Distill Qwen-7B (Distill) | **78.2** | **79.2** | **73.7** | **84.5** | **87.2** | **83.1** | 76.0 | **90.2** | **84.2** |

## 5.2 Result and Analysis

### 5.2.1 Benchmark Performance

As illustrated in Figure 3, for the finetuned models in the first row, the MAoP demonstrates a substantial performance enhancement over the long-horizon thinking (RL w/ *Long/Artifact.*) baselines, achieving a remarkable 5% to 40% improvement across all planners. Although trained with the same dataset, MAoP achieves higher scores than RL w/ *Long/Artifact.* in CPL by better constraint compliance and CPH by integrating more details from the long context to the plan. When compared with naive wide-horizon thinking (Zero-shot w/ *Wide/Aritifact.*), MAoP excels especially in stronger strategists, as evidenced by higher FEA and PER scores, suggesting that the strategist plays an important role in the planning process with performance better than artifactual guidance. For advanced models as zero-shot planners, e.g., Deepseek-R1 and Gemini-2.5, the strategist also significantly boosts the performance in travel planning compared to zero-shot baselines.

### 5.2.2 Inference-Time Scaling Capability of the Strategist

As shown in Figure 4, we find that strategists can consistently improve the performance by scaling up more considered aspects. We find that even if the model size is the same, the stronger strategist with thinking (R1-Distill 7B) has better scalability,

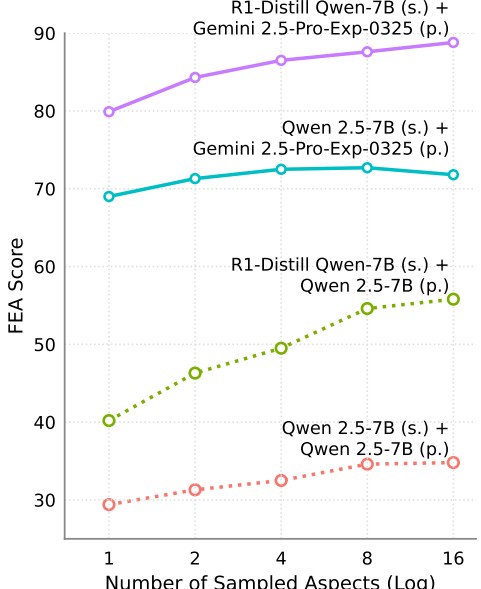

Figure 4: We experiment two strategists (Qwen 7B & R1-Distill Qwen-7B) and two planners (Qwen 7B & Gemini 2.5) to showcase the scaling capability of the strategists.

especially for advanced planners like Gemini 2.5. In contrast, the Qwen 2.5-7B model has limited scalability, because it cannot effectively route a suitable planning blueprint when dealing with increasing aspects.

### 5.2.3 MAoP Distillation

As shown in Table 3, distilling *R1-Distill 7B (s.) + Gemini 2.5-Pro-Exp-0325 (p.)* enables 3B sized models to outperform the MAoP combinations even with larger model sizes. For R1-Distill 7B with the thinking mode, we also distill the thinking part of planning from the Gemini. Compared to the MAoP combination of the same models, R1-Distill 7B (Distill) achieves better performance with distilled one-step wide-horizon thinking. These results demonstrate that distillation from advanced models to smaller models achieves substantial performance improvements even without the original multi-turn MAoP process. The enhancement becomes increasingly pronounced as the capability gap between teacher and student models widens, even surpassing the MAoP training models.

### 5.2.4 Emergence of Spontaneous Behaviors in Causal Travel Simulation

Given the causal nature of an itinerary, the traveler agent exhibits emergent behavior by spontaneously adjusting its predetermined plan in response to unfolding events. From trajectories in Figure 5, this elderly couple chooses not to have dinner in Nanmen Lamb Hot Pot because they are too tired to travel to another place, especially since they just suffered from a long train trip to Beijing this morning. This indicates that neglecting causal dependencies in real-world planning evaluation can lead to significant deviations from the real-world situation.

**A 3-day trip to Beijing**

An elderly couple who love culture and history and prefer a relaxed way of traveling. They enjoy visiting temples and savoring local cuisine, while also emphasizing rest and comfort during their journey.

**Trajectory from plan**

Day 1:
10:00 Beijing West Railway Station
11:00 Kapok Hotel
12:30 Siji Minfu Restaurant
14:30 Shichahai
18:00 Nanmen Lamb Hot Pot
20:00 Kapok Hotel

*The elder couple skips this activity in simulation due to the exhaustion of traveling to Beijing and sightseeing.*

Day 2:
08:00 Kapok Hotel
09:00 The Palace Museum (Forbidden City)
12:30 The Corner Tower Café (Forbidden City)
14:30 Jingshan Park
17:30 Donglaishun Restaurant
19:30 Kapok Hotel

Day 3:
08:00 Kapok Hotel
09:30 Yonghe Temple (Lama Temple)
12:00 Huguo Temple Snacks
13:30 Kapok Hotel
14:00 Beijing South Railway Station
15:00 Beijing South Railway Station

**Trajectory from simulation**

Day 1:
10:00 Beijing West Railway Station
10:38 Kapok Hotel
12:04 Siji Minfu Restaurant
14:02 Shichahai
18:38 Kapok Hotel

Day 2:
08:00 Kapok Hotel
08:09 The Palace Museum (Forbidden City)
12:39 The Corner Tower Café (Forbidden City)
14:11 Jingshan Park
16:06 Donglaishun Restaurant
18:06 Kapok Hotel

Day 3:
08:00 Kapok Hotel
09:34 Yonghe Temple (Lama Temple)
11:31 Huguo Temple Snacks
14:00 Kapok Hotel
14:40 Beijing South Railway Station

Figure 5: The simulation trajectory is not always consistent with the planned one because the traveler agent can change the subsequent itinerary based on the current situation. The **FEA** score is used to calculate the similarity of these two trajectories.

## 6 Related Work

**LLM Planning** The planning capabilities of LLMs have become a key research focus due to their potential as autonomous problem-solving agents [25, 7, 26]. Recent studies have advanced these capabilities through developments in task decomposition, multi-step reasoning [17, 27], and adaptive planning [28, 29]. While these planning algorithms have shown promising results [30, 31], their planning scenarios are limited to simple tasks with a single objective function. For complex real-world planning like travel planning, previous researchers focus on how to preprocess the various information and constraints to make LLM easier to understand, but few of them explore how to enhance the planning ability in this complex scenario.

**Generative Simulation** LLM agents begin to exhibit strong capabilities in mimicking human behaviors [32, 33]. Some researchers [34, 35, 36] have been investigating the behavioral patterns of human-like LLM agents within sandbox environments, focusing on simulating human social interactions and lifestyles to study their social behaviors. Although simulation-based evaluation has become a prevalent methodology in robotics research [37, 38], our work represents the comprehen-

sive investigation into evaluating complex task performance using LLM agents within simulated environments.

# 7 Conclusion

We propose MAoP to enhance wide-horizon thinking for solving real-world planning problems with multifaceted constraints. In addition, we propose Travel-Sim, an evaluation benchmark that leverages agent-based simulation to offer causal and multi-granularity evaluation in travel planning scenarios. Our contributions advance the wide-horizon thinking capabilities of LLMs in real-world planning and offer novel insights for evaluating sophisticated scenarios through agent-based simulation.

## Acknowledgement

Dongjie Yang and Hai Zhao are with the Department of Computer Science and Engineering, Shanghai Jiao Tong University; Key Laboratory of Shanghai Education Commission for Intelligent Interaction and Cognitive Engineering, Shanghai Jiao Tong University; Shanghai Key Laboratory of Trusted Data Circulation and Governance in Web3.

This paper was completed during Dongjie Yang's internship at Xiaohongshu Inc. and was supported by the Joint Research Project of Yangtze River Delta Science and Technology Innovation Community (No. 2022CSJGG1400).

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

## Appendix Contents

# A Limitations and Ethics Statement

## A.1 Limitations

While our proposed method (MAoP) and agent-based simulation benchmark (Travel-Sim) demonstrate significant advancements in complex travel planning and evaluation, several limitations warrant discussion.

### A.1.1 MAoP Limitations

Though MAoP significantly enhances wide-horizon thinking with deliberate pre-planning by the strategist, MAoP still exhibits a few limitations: 1) MAoP requires the collaboration between the strategist and planner, and the conduction of aspect-aware thinking in multiple turns of dialogue. This complicated process costs more inference time than directly using CoT. To alleviate this, we propose that the MAoP distillation can compress this complicated process, gaining similar efficiency compared to CoT. 2) We find that weak strategists, especially those with sizes smaller than 7B, can cause severe performance degradation due to low-quality planning blueprint and poor inference-time scalability. The recommended way to enhance wide-horizon thinking ability for smaller models is to distill the outputs of the advanced model after MAoP training.

### A.1.2 Travel-Sim Limitations

The Travel-Sim is the pioneer in evaluating with agent-based simulation, offering a novel solution for complex scenario evaluation, e.g., travel planning. Agent-based simulation and evaluation have several limitations that can be discussed: 1) Similar to other LLM-based evaluation benchmarks, the "traveler", which is an LLM, has the potential bias and unreproducibility over the evaluation. To mitigate this limitation, we stipulate the agent model to be Gemini-2.5-Pro-Exp-0325 [21] using greedy decoding (0 temperature) for reproducibility. We also verify the consistency between LLM-based and human-based evaluation in the Appendix F.5, showcasing the 92% consistency in PER score. 2) Although Travel-Sim takes causality and real-world information into account, we do not include all the emergencies that may happen in the real trip, especially some force majeure and human factors. For example, bad weather or delays caused by carelessness also significantly influence the trip. 3) Although Travel-Sim offers causal and multi-granularity evaluation, the whole evaluation process is time-consuming and costs about 20K tokens per sample. To evaluate the 112 samples of the entire dataset, we cost around $12 by consuming about 2M tokens in total.

## A.2 Ethics Statement

For the traveler profiles in the training dataset and Travel-Sim, we do not use any personal information to collect traveler profiles. The diversity of travelers is manually designed, and data is synthesized using Gemini-2.0-Pro-Exp-0205 [21]. For the sightseeing event agent that uses travel blog posts to simulate travel experience, we collect blog posts from Red Note (Xiaohongshu) that may contain personal information and copyrighted items. Therefore, people using the blog posts should respect the privacy and copyrights of the blog post owner and strictly agree to the license in Appendix B.

# B License

By downloading or using our open-source data, you understand, acknowledge, and agree to all the terms in the following agreement.

**ACADEMIC USE ONLY** Any content from the Travel-Sim dataset is available for academic research purposes only. You agree not to reproduce, duplicate, copy, trade, or exploit for any commercial purposes.

**NO DISTRIBUTION** Respect the privacy of the personal information of the original source. Without the permission of the copyright owner, you are not allowed to perform any form of broadcasting, modification, or any other similar behavior to the data set content.

**RESTRICTION AND LIMITATION OF LIABILITY** In no event shall we be liable for any other damages whatsoever arising out of the use of, or inability to use this dataset and its associated software, even if we have been advised of the possibility of such damages.

**DISCLAIMER** You are solely responsible for legal liability arising from your improper use of the dataset content. We reserve the right to terminate your access to the dataset at any time. You should delete the Travel-Sim dataset if required.

You must comply with all terms and conditions of these original licenses, including but not limited to the Google Gemini Terms of Use, the Copyright Rules & Policies of Red Note (Xiaohongshu). This project does not impose any additional constraints beyond those stipulated in the original licenses.

## C   Preprocessing Framework

In this section, we introduce how to collect and preprocess the heterogeneous information as the "long context" for travel planning. **This context information serves as the information source input for both the training dataset and Travel-Sim.**

We break the information and constraints necessary for planning into different categories shown in Figure 6. We leverage our framework to collect and preprocess them for further planning. The framework consists of multiple modules that collaborate with each other.

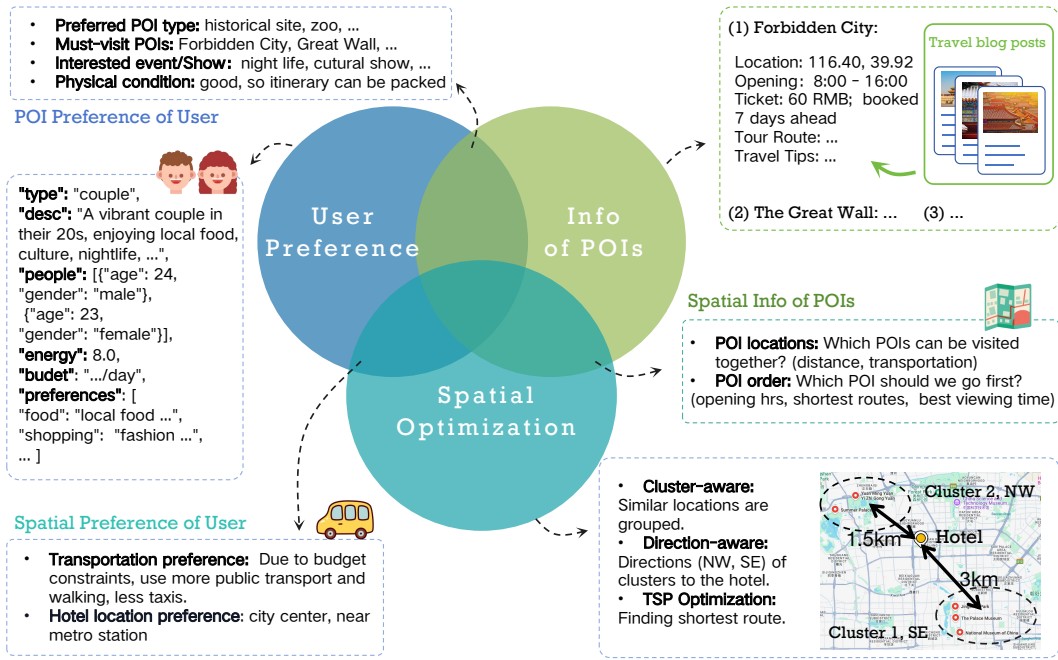

Figure 6: The decomposition of multifaceted constraints and heterogeneous information to be considered when creating a travel plan.

### C.1   Proactive Consultant Agent for Implicit Preferences Elicitation

Users often provide insufficient information, making it challenging for the assistant to devise an effective plan. To address this, we not only extract preferences from user requests but also proactively ask questions to elicit implicit preferences. We design two agents to gather users' travel and accommodation preferences, respectively. The travel consultant agent collects basic trip details (group size, duration, traveler type) and integrates insights from travel blogs to identify potential challenges for travelers. The accommodation consultant first inquires about hotel preferences and then searches travel blogs and maps to identify optimal hotels. Through iterative interactions, agents can uncover implicit preferences for travel planning and accommodation.

### C.1.1 Travel Consultant Agent

We develop an LLM-based agent to serve as a travel consultant, designed to comprehensively dig out the user's implicit preferences. We design six aspects that the travel consultant agent needs to take into account, as shown in Table 4.

Table 4: Considerations for implicit preferences elicitation.

| Type | Description |
|---|---|
| Duration of Visit | How long does the user wish to tour, is it one day, two days, or longer? |
| Group Composition | How many people are in the group, what is their approximate age, and are there any special groups, such as the elderly or children? |
| Attraction Type | What type of attractions does the user hope to visit, cultural relics, natural scenery, or shopping and entertainment? |
| Number of Attractions | Does the user want to visit as many attractions as possible, or not too many? |
| Mode of Transportation | Under replaceable circumstances, what mode of transportation does the user prefer: walking, cycling, public transport, taxi, or driving? |
| Special Requirements | Does the user have any special needs or preferences, such as particular requirements for shopping or sightseeing? |

Based on the initial user request, the travel consultant agent will proactively inquire about several questions concerning six aspects to better understand the traveler's situation. Upon gathering sufficient information across these six aspects, the agent will synthesize a comprehensive conclusion in the form of a traveler profile. The agent will subsequently present this profile to the user for review, allowing them to verify the details and provide any necessary additions or corrections through further interaction with the agent.

Instead of asking the same fixed questions, travel consultant agents have the following advantages: 1) Customized Service: we observe that the LLMs will adjust the questions based on the destination and the traveler type, and also give relevant advice to guide the user. 2) Feedback: The agent can receive feedback from the user and raise further interaction to refine the traveler profile.

### C.1.2 Hotel Consultant Agent

The selection of the hotel is the key part of travel planning, which previous researchers seldom take into account. The hotel influences not only the accommodation experience but also the planning and logistics of the entire trip, concerning the hotel locations. We develop an LLM-based agent to serve as a hotel consultant agent to recommend hotels based on travelers' preferences.

We decompose this process into five phases: 1) **Inquiry phase**: similar to the travel consultant agent, the hotel consultant agent has to inquire about the user preferences. 2) **Online searching phase**: after understanding the hotel and POI preferences, the agent can search the internet to find the compliant hotels from the latest tour guides. 3) **Map searching phase**: the agent leverages the map API to search for hotels near the most-visited locations and conclude a list that meets the requirements. 4) **Conclusion phase**: the agent recommends the hotels from the travel blog posts and the map API. 5) **Feedback phase**: the agent receives the feedback and revises the recommendation.

To be specific, in the inquiry phase, we consider four aspects, as shown in Table 5. After several proactive interactions, the agent will stop asking if it has collected enough user preferences. Besides the hotel preference, the hotel agent can leverage POI preferences and spatial information (POI clusters) from other modules. In the online searching phase, we use the search API provided by the Red Note (Xiaohongshu) app, a popular Chinese app focused on lifestyle content sharing. We use the Red Note (Xiaohongshu) search API to search for blog posts related to hotel recommendations. In the map searching phase, we use the Amap (Gaode) Map API, a Chinese map app similar to Google Maps, to search for hotels near the most-visited locations. The probable most-visited locations are predicted by the agent based on the given information. In the conclusion phase, the agent will recommend the hotels provided by the blog posts and the map API by comprehensively taking the locations and preferences into account.

Table 5: Considerations for accommodation planning.

| Type | Description |
|------|-------------|
| Hotel Consistency | Does the user prefer to stay in the same hotel throughout the trip, or change hotels according to the attractions? |
| Accommodation Budget | What is the user's budget for accommodation? |
| Location Preference | Does the user have any requirements for the location of the accommodation, such as being in the city center, near attractions, etc.? |
| Facility Requirements | What facilities does the user require from the accommodation? |

## C.2 Preference-aware POI Selection

As we have the user preferences collected by the travel consultant agent, we leverage the information to find the potential POIs and activities that will appeal to users. We first leverage LLMs to generate several search queries based on user preferences and use the Red Note (Xiaohongshu) API to search the relevant blog posts. Based on the preferences, we use LLMs to extract the POIs that match user preferences. We use the Amap (Gaode) map API to obtain the POI meta information. We finally filter and deduplicate them to get a list of candidate POIs.

## C.3 Cluster-based Spatial Optimization

To enhance the coherence of travelers' journeys and minimize unnecessary back-and-forth travel, similar to ITINERA [23], we cluster the POIs based on geographical distances from each other. This way, visitors can follow a well-planned, spatially coherent route for their tours, for example, visiting the POIs in the same cluster in one day, to enjoy a travel experience that is both efficient and pleasant. We use K-means++ to cluster the POIs, where the number of clusters is determined jointly by the number of travel days and the candidate POIs.

After obtaining the POI clusters, we can describe the spatial distribution characteristics of POIs from two perspectives: intra-cluster features and inter-cluster relationships, to enable LLMs to understand the spatial information.

---

**Algorithm 1** Find Shortest Route in the POI Cluster

---

**Input:** locations set $L$, start location name $s$, end location name $e$
**Output:** shortest path $P$, shortest distance $d_{total}$, step distances $\{d_i\}_{i=1}^{n-1}$
**function** FINDSHORTESTROUTE($L, s, e$)
$\quad$ Extract other locations excluding $s$ and $e$: $O \leftarrow \{loc \in L \mid loc \notin \{s, e\}\}$
$\quad$ Let $n \leftarrow |O|$
$\quad$ **if** $n < 6$ **then**
$\quad\quad$ Compute all permutations of $O$: $\Pi \leftarrow \{\pi \mid \pi \text{ is a permutation of } O\}$
$\quad\quad$ Find shortest path and distance from $\Pi$:
$(P, d_{total}) \leftarrow \arg\min_{\pi \in \Pi} \sum_{i=1}^{|\pi|} \text{CalculateDistance}(C[\pi[i]], C[\pi[i+1]])$
$\quad$ **else**
$\quad\quad$ Get initial path by finding nearest neighbors: $P \leftarrow \text{NearestNeighbor}(L, s, e)$
$\quad\quad$ Optimize path using 2-opt search: $P \leftarrow \text{TwoOptSearch}(P, C, s, e)$
$\quad\quad$ Calculate shortest distance: $d_{total} \leftarrow \sum_{i=1}^{|P|-1} \text{CalculateDistance}(C[P[i]], C[P[i+1]])$
$\quad$ **end if**
$\quad$ Calculate step distances: $\{d_i\}_{i=1}^{|P|-1} \leftarrow \{\text{CalculateDistance}(C[P[i]], C[P[i+1]])\}_{i=1}^{|P|-1}$
$\quad$ **return** $(P, d_{total}, \{d_i\}_{i=1}^{|P|-1})$
**end function**

---

### C.3.1 Intra-cluster: Finding the Shortest Routes within the Cluster

Since the POIs in the same cluster are relatively close to each other, travelers have the potential to visit them in sequence within a single day. In order to find the shortest route that does not retrace

any part of the path, this problem can be formulated as a variant of the Traveling Salesman Problem (TSP). As the classic Traveling Salesman Problem (TSP) is an NP-hard problem in combinatorial optimization, we use brute force to solve clusters with a small number of POIs (fewer than 6), and for those with a larger number, we use the 2-opt algorithm to find an approximate solution. As shown in Algorithm 1, we have the start location $s$ and end location $e$, which are generally set as the hotel. Locations besides $s$ and $e$ are the POIs in the same clusters. We calculate the shortest route that starts at the hotel and ends at the hotel after visiting the POIs.

### C.3.2 Inter-cluster: Relative Cluster Locations to the Hotel

The locations of the POI clusters relative to hotels are also key information for LLM to know the spatial distribution of all POIs. We mainly calculate the direction and distance from the cluster center to the hotel, as shown in Algorithm 2. To make directions easier to understand for LLMs, we can simplify directions into eight basic compass points.

---

**Algorithm 2** Calculate Directions and Distances

---

**Input:** Start point $S = (\lambda_s, \phi_s)$, set of target points $\mathcal{T} = \{(T_i, C_i)\}_{i=1}^n$ where $C_i = (\lambda_i, \phi_i)$
**Output:** Set of tuples $\mathcal{O}$ containing direction $D_i$ and distance $d_i$ for each target point $T_i$
**function** CALCULATEDIRECTIONSANDDISTANCES$(S, \mathcal{T})$
    Initialize empty set $\mathcal{O}$
    **for all** $(T_i, C_i) \in \mathcal{T}$ **do**
        Convert coordinates to radians: $\lambda'_s, \phi'_s, \lambda'_i, \phi'_i$
        Calculate geodesic distance $d_i$ between $S$ and $C_i$
        Compute bearing $\theta'$:

$$\theta' = \arctan 2 \left( \sin(\lambda'_i - \lambda'_s) \cdot \cos(\phi'_i), \cos(\phi'_s) \cdot \sin(\phi'_i) - \sin(\phi'_s) \cdot \cos(\phi'_i) \cdot \cos(\lambda'_i - \lambda'_s) \right)$$

        Normalize bearing $\theta$: $\theta = (\theta' \cdot \frac{180}{\pi} + 360) \mod 360$
        Map bearing to cardinal direction $D_i$: $D_i = \text{directions}[\text{round}(\theta/45) \mod 8]$
        Add tuple $(D_i, d_i)$ to $\mathcal{O}$
    **end for**
    **return** $\mathcal{O}$
**end function**

---

### C.4 Entire Preprocessing Workflow

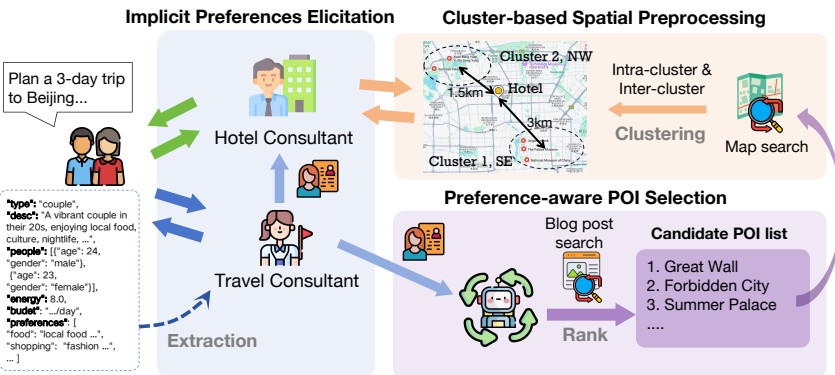

Figure 7: The entire workflow of the preprocessing framework.

**Workflow** The preprocessing framework consists of these modules above that work together in collaboration rather than operating independently. As shown in Figure 7, we present the entire workflow of the framework. First, as the user raises a request for planning a trip, the travel consultant agent gathers user preferences and information by interacting with the user. Based on the preferences, we search the relevant blog posts and extract the candidate POIs. We perform clustering on the candidate POIs. In the next step, the hotel consultant inquires about the hotel preference and

comprehensively considers all the information (user profile, hotel preference, and spatial information) to recommend hotels. After we have the location of the hotels, we conduct intra-cluster and inter-cluster analysis.

Table 6: The composition of the long context input.

| Type | Source | Description |
|------|--------|-------------|
| User Profile & Preferences | Travel Consultant | The information of the traveler/traveler group and corresponding preferences. |
| Hotel Information | Hotel Consultant | The location and other information about where the user lives. |
| POI Blog Posts | Blog Post Search | The relevant blog posts that provide sightseeing tips for each POI. |
| Inter-Cluster Information | Spatial Preprocessing | The shortest routes within the POI cluster. |
| Inter-Cluster Information | Spatial Preprocessing | The relative direction and distance from cluster centers to hotels. |

**Final Input Context for LLMs**    After we collect and preprocess the information, we organize it as the context part of the LLM Input. To give an intuitive view, we structure the context in Table 6. It is noted that for every candidate POI, we search for the most relevant blog post that provides the latest sightseeing tips. We can not depend on the intrinsic knowledge of LLMs about the POIs, which is outdated and inaccurate most of the time. The latest blog posts provide most of the information and tips for visiting the POI, for example, the optimal sightseeing routes, which help LLMs to optimize the itinerary and personalize the plan.

## D    Preliminary Experiment Details

### D.1    Detailed Setup

**Naive Wide-Horizon Thinking with Artifactual Guidance**    In the preliminary experiment of exploring naive wide-horizon thinking, we craft the artificial guidance that includes a series of aspects to be considered. We manually design five aspects as shown below:

> ➢ **Aspect I: Analyze Key Information in the Traveler's Question and Extract Their Needs**
> Regional Scope (e.g., Beijing, Shanghai Inner Ring): ...
> Time Range (duration of the trip): ...
> Group Composition (couples, families, etc., and the number of people): ...
> Special Preferences/Requirements (specific locations, events, travel style, etc.): ...
>
> ➢ **Aspect II: Analyze the POI Selection and Combination**
> Interests and Hobbies: Select suitable tourist attractions based on the travelers' preferences and group composition. Are there any specific tourist attractions the travelers are particularly interested in?
> Time Requirements: Are there any specific time constraints (e.g., opening hours) for visiting the tourist attractions? Is there an optimal time for sightseeing?
> Duration Requirements: How long will each POI visit take? Can the duration be adjusted flexibly according to the travelers' preferences?
> Special Experiences: Are there any unique activities or experiences available? Any performances or special events?
>
> ➢ **Aspect III: Analyze the Route Planning**
> Tourist Route: Based on [Spatial Optimization Analysis], is there a more efficient, logical route that minimizes backtracking?
> Attraction Locations: Take the travelers' accommodation into account when planning the distances and directions between tourist attractions. For more distant attractions, should transportation time be considered? Can nearby attractions be grouped for the same day?
> Transportation Suggestions: Offer suitable transportation recommendations based on the group type and route plan. Ensure transportation time is factored into the schedule.
>
> ➢ **Aspect IV: Analyze the Itinerary's Comfort Level**
> Travel Pace: Adjust the pace according to the group demographics and physical stamina. Balance sightseeing, rest periods, and free time to ensure a pleasant and manageable experience.
> POI Scheduling: Plan the number of attractions visited per day according to the travelers' interests and energy levels. Avoid overly packed schedules that lead to fatigue or overly sparse plans that risk boredom.
> Dining Recommendations: Schedule meal times thoughtfully and provide dining suggestions. Good meal planning ensures travelers remain energized and comfortable, avoiding exhaustion or hunger.
>
> ➢ **Aspect V: Analyze Travel Strategies and Precautions**
> Ticket Purchase: Determine whether tickets need to be bought in advance, the methods for purchasing, ticket prices, discounts, and whether reservations are required.
> Scenic Spot Guidelines: Offer tips and precautions to avoid potential inconveniences. Provide recommended routes and suggested sequences for touring the scenic spots.

**Naive Wide-Horizon Thinking with Self-Generated Guidance**    Instead of using the artifactual guidance, we leverage the LLM to first analyze the aspects to be considered and generate the corresponding guidance. This process can be perceived as the zero-shot strategist, as shown below:

```
------[Available Information (Start)]------
{information}
------[Available Information (End)]------
------[User Instruction (Start)]------
{instruction}
-------[User Instruction (End)]------

You need to analyze the instruction in [User Instruction] with considering the [Available Information], and break down the user's instruction
into multiple sub-steps based on different aspects. You do not need to fulfill the request of the instruction in [User Instruction], but only
provide various aspects and suggestions that help address the instruction. For each thinking aspect, you should provide detailed guidance
to facilitate more specific thinking in the next step, and attempt to refer information from [Available Information].
Additionally, you should prioritize these aspects in accordance with the thinking process (starting with thinking aspect 1, followed by
thinking aspect 2, and so on). Therefore, this coherence thinking process will more effectively address the user's instruction.

Reply in Markdown format:
## (Aspect 1)...
......(Provide guidance + quote information, no need to address the user instruction)
## (Aspect 2)...
......(Provide guidance + quote information, no need to address the user instruction)
```

## D.2   Scaling Capability of Naive Wide-horizon Thinking

We investigate the potential benefit of scaling inference-time computation with more aspects for naive wide-horizon thinking.

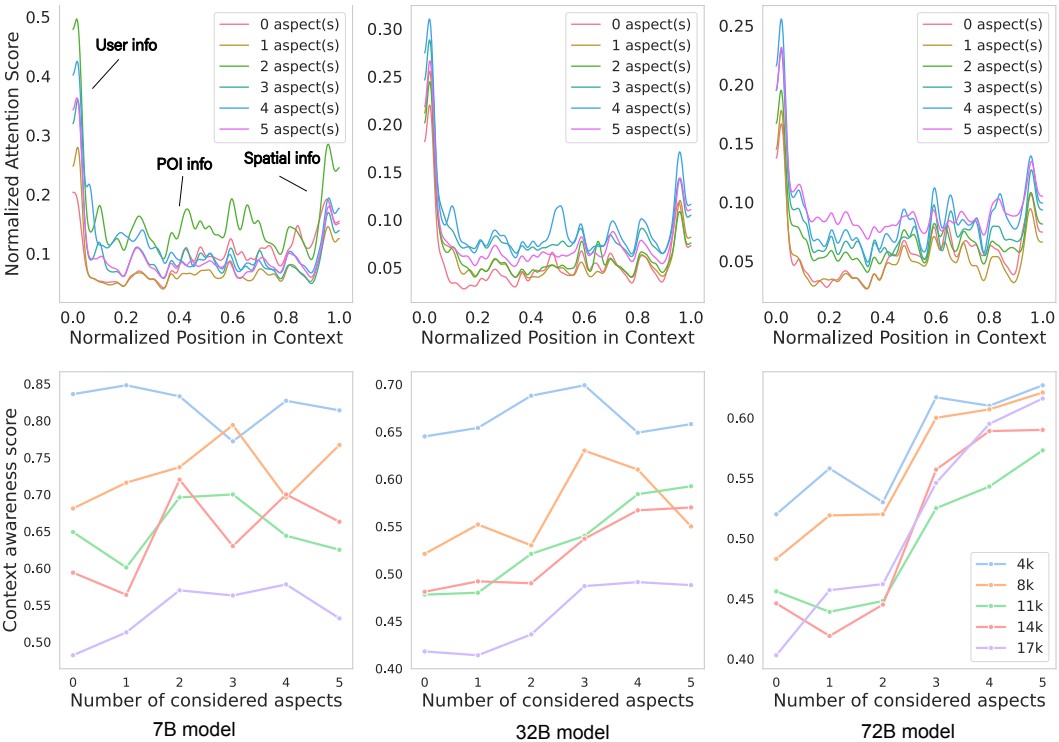

Figure 8: When LLMs consider different numbers of aspects, we analyze the attention pattern of the context (excluding the instruction) on different sizes of Qwen-2.5, varying from 7B, 32B, and 72B (from left to right). **(Upper row)** We present the distribution of attention scores across the context, aggregated from the output tokens. **(Lower row)** Additionally, we investigate the impact of context length (number of context tokens) by examining the context awareness scores. Our analysis reveals that larger models exhibit an enhanced capability for wide-horizon thinking, enabling them to focus on a broader range of information within longer contexts by considering multiple aspects simultaneously.

| Aspect | Qwen 2.5-7B | | | | Qwen 2.5-32B | | | | Qwen 2.5-72B | | | |
|---|---|---|---|---|---|---|---|---|---|---|---|---|
| | CPH ↑ | CPL ↑ | FEA ↑ | PER ↑ | CPH | CPL | FEA | PER | CPH | CPL | FEA | PER |
| 0 | 18.2 | **40.2** | 15.2 | 27.8 | 39.6 | 51.1 | 21.7 | 34.6 | 50.4 | 62.5 | 37.0 | 44.0 |
| 1 | 20.3 | 39.5 | 14.4 | 28.5 | 42.8 | **53.3** | 25.4 | 33.3 | 51.4 | 60.3 | 37.2 | 43.4 |
| 2 | 25.3 | 36.2 | **19.0** | 30.6 | 47.0 | 50.1 | 31.4 | 38.7 | 54.9 | 60.2 | 39.2 | 46.0 |
| 3 | **31.5** | 37.3 | 16.3 | **32.6** | 49.9 | 50.4 | **33.0** | 42.3 | 56.5 | **62.9** | 41.5 | 49.7 |
| 4 | 29.8 | 37.8 | 14.5 | 29.4 | **52.9** | 47.2 | 32.0 | **43.5** | 60.0 | 61.7 | 41.8 | 52.5 |
| 5 | 28.3 | 38.0 | 17.8 | 27.7 | 52.7 | 48.8 | 31.9 | 43.1 | **60.3** | 59.5 | **42.1** | **53.5** |

Table 7: Comparing the effectiveness of wide-horizon thinking when considering a different number of aspects by quantitative metrics.

**Experiment Setup**    We investigate scenarios ranging from 0 aspect (i.e., zero-shot CoT baseline without artifactual guidance) to a maximum of 5 artifactual aspects considered. We evaluate how effectively context is considered in the thinking process by analyzing attention patterns with respect to context. Besides attention patterns, we evaluate if wide-horizon thinking is effective and enhances the quality of travel plans by leveraging four metrics: CPL, CPH, FEA, and PER scores. We conduct the experiments on Qwen-2.5 [19] with different sizes, including 7B, 32B, and 72B.

**Result Analysis**    In the upper part of Figure 8, we show curves of the attention score distributions over the context (excluding the instruction). The attention scores are aggregated from the output tokens and then averaged across layers and attention heads. We observe that larger LLMs (34B and 72B) exhibit higher attention scores across the context when considering more aspects, indicating a strong ability to focus on relevant details. In contrast, the smaller LLM (7B) does not follow this pattern, achieving its highest scores when two aspects are considered. In addition, we introduce another indicator, dubbed context awareness score $\mathcal{S}_c$, to indicate the proportion of the most attended tokens in the context out of the total, as calculated in Equation 2:

$$\mathcal{S}_c = \frac{1}{n} \sum_{i=0}^{n} \mathbb{I}_{\mathcal{A}^c \in \text{Top}k}(\mathcal{A}_i^c), \tag{2}$$

where $A^c$ denotes the attention scores over the context from the output tokens. In the lower part of Figure 8, for larger models like 72B, as the context length increases, the $\mathcal{S}_c$ curves of longer length rise more steeply. It indicates that by considering more aspects, LLMs can capture finer details in longer contexts. This phenomenon emerges only when models are large enough, as smaller LLMs do not exhibit explicit improvements in $\mathcal{S}_c$ with long contexts.

As shown in Table 7, larger LLMs demonstrate enhanced travel planning capabilities as they consider more aspects, achieving progressively higher FEA and PER scores. However, this improved performance comes at the cost of formatting compliance, as evidenced by decreasing CPL scores.

# E    MAoP Experiment Details

## E.1    Training Details

### E.1.1    Reward Model

**Training Dataset**    As shown in Table 9, we collect 14 diverse cities as destinations and 37 traveler types. Based on these destinations and travelers, we first synthesize 1K travel planning requests. We then separately use Gemini-2.0-Pro-Exp-0205 and Qwen 2.5-32B to directly generate travel plans. For about 2K generated plans, we conduct the simulation-based evaluation to get the PER scores as the reward score labels.

**Model Structure**    To avoid reward hacking, instead of directly predicting the reward (PER-agg. score), the reward model is required to generate the corresponding scores for the five criteria in the PER score and aggregate them into the final reward.

**Training Setup**    We conduct the pointwise generative reward modeling. We use the travel plan as the input. Instead of directly using the aggregated scalar score calculated in Equation 3, we extract the assessment text of the traveler and the corresponding PER score after the whole journey as the output. We use 8 H800 GPUs to finetune the Qwen 2.5-7B model for 5 epochs with a learning rate of 2e-5, global batch size of 64.

### E.1.2   Strategist

**Training Dataset**    To train a strategist with RFT, we construct the training dataset consisting of 50K samples with 120 distinct traveler types and 20 Chinese cities. We first conduct RFT on Deepseek-R1-Distill Qwen-7B to get the initial RFT data. We reject the sample that gets the average PER score lower than 40 after sampling $N = 3$ times. We use 32 H800 GPUs to train Deepseek-Distill Qwen-7B for 3 epochs with a learning rate of 2e-5, global batch size of 128.

**Training Setup**    After training the Deepseek-R1-Distill Qwen-7B, we finally collect 22K RFT data with rejection sampling outputs. We reuse this 22K data to finetune the Qwen 2.5 7B and 32B. We use 8 H800 GPUs to train Qwen 2.5 7B for 3 epochs with a learning rate of 2e-5, global batch size of 64. We use 32 H800 GPUs to train the Qwen 2.5-32B strategist for 3 epochs with a learning rate of 1e-5, global batch size of 128.

### E.1.3   Planner

**Training Dataset**    We first train the planner using the 22K data above as the cold start for subsequent RL training. To train the planner with RL, we additionally construct the training dataset consisting of 20K samples.

**Training Setup**    We conduct the standard GRPO [11] using the veRL [39] framework. We use 32 H800 GPUs to train Deepseek-R1-Distill Qwen-7B, Qwen 2.5-32B, and Qwen 2.5-7B, for 200 steps with learning rates of 1e-6, 5e-7, 1e-6, respectively, and the KL loss coefficient of 0.001. We use the KL penalty to prevent severe biases. For loss calculation, we mask the aspect-aware guidance from the strategists, preventing the planners from generating the guidance by themselves.

### E.1.4   Distillation

**Training Dataset**    We reuse the input of the 22K data and use *Deepseek-R1-Distill Qwen-7B (s.) + Gemini-2.5-Pro-Exp-0325 (p.)* to generate the output. For each sample, we sample the output 3 times. We use the reward model to filter out the output whose reward score is lower than 80. We use the filtered dataset consisting of 15K samples to train the Qwen 2.5-3B, Llama 3.2-3B, and Deepseek-R1-Distill Qwen-7B. For the first two smaller models, we discard the thinking part. For Deepseek-R1-Distill Qwen-7B, we use the thinking part from Gemini-2.5-Pro-Exp-0325.

**Training Setup**    We use 32 H800 GPUs to train Deepseek-Distill Qwen-7B for 5 epochs with a learning rate of 2e-5, global batch size of 128. We use 8 H800 GPUs to train Qwen 2.5 3B and Llama 3.2 3B for 5 epochs with a learning rate of 2e-5, global batch size of 64.

## E.2   Inference Details

### E.2.1   Inference for Strategist

**Decomposition**    The first stage of pre-planning is to decompose the planning request into multiple aspects to be considered and provide guidance for further analysis. Here is the prompt used in the RFT and inference:

```
------[Available Information (Start)]------
{information}
------[Available Information (End)]------
------[User Instruction (Start)]------
{instruction}
------[User Instruction (End)]------

You need to analyze the instruction in [User Instruction] with considering the [Available Information], and break down the user's instruction
into multiple different aspects to be considered. You do not need to fulfill the request of the instruction in [User Instruction], but only
provide various aspects and suggestions that help address the instruction. For each aspect, you should provide detailed guidance to
facilitate more specific thinking in the next step, and attempt to refer information from [Available Information].

Reply in Markdown format:
## (Aspect 1)...
......(Provide guidance for further thinking + quote information, no need to address the user instruction)
## (Thinking Aspect 2)...
......(Provide guidance for further thinking + quote information, no need to address the user instruction)
```

**Routing**    If we apply these aspects for subsequent analysis following the naive wide-horizon
paradigm, the planner's performance will exhibit severe scalability limitations as the number of
aspects increases.  Therefore, the strategist reorganizes these aspects and integrates them into a
coherent planning blueprint. Here is the prompt used in the RFT and inference:

```
------[Available Information (Start)]------
{information}
------[Available Information (End)]------
------[User Instruction (Start)]------
{instruction}
------[User Instruction (End)]------
------[Aspect Reference for Further Analysis (Start)]------
{aspect}
------[Aspect Reference for Further Analysis  (End)]------

We have analyzed the instruction in [User Instruction] with considering the [Available Information], and break down the user's instruction
into multiple different aspects for further analysis. For each aspect, we provide detailed guidance to facilitate more specific thinking in the
next step, and attempt to refer information from [Available Information].

However, the aforementioned aspects for further analysis may involve overlapping or fragmented parts. Your task is to reorganize them
into a unified planning blueprint:
1. Distill exclusively instruction-oriented perspectives and corresponding guidance by consolidating redundant aspects into more refined
analytical dimensions.
2. Develop a systematic planning blueprint where each subsequent aspect logically incorporates insights from preceding aspects. You
should prioritize these aspects in accordance with the planning process (starting with aspect 1 + guidance, followed by aspect 2 +
guidance, and so on). You need to provide coherence planning blueprint that more effectively helps the planning model to further conduct
in-depth anaylysis to address the user's instruction, so **you do not need to directly address the user instruction**.

Reply in Markdown format:
## First, (Aspect 1)...
......(Provide guidance + quote information, no need to address the user instruction)
## Secondly, (Aspect 2 based on previous context)...
......(Provide guidance + quote information, no need to address the user instruction)
```

### E.2.2    Inference for Planner

**Aspect-Aware Thinking**    Based on the planning blueprint, the planner conducts a more focused and
in-depth analysis following the order in the blueprint. The whole process of aspect-aware thinking
is implemented through multiple turns of dialogue with shared history. We instruct the planner to
follow the guidance to analyze one of the aspects in a turn of dialogue, as shown below:

```
------[Aspect Reference for Further Analysis (Start)]------
{one of the aspects}
------[Aspect Reference for Further Analysis  (End)]------

Based on the provided information and previous analysis, follow [Aspect Reference for Further Analysis] to deliver an enhanced detailed
analysis from that aspect.
```

**Formatting Output Plan**    To integrate the previous analysis, we design formatting instructions for
the model to output in a specified format. You can refer to the case study in Section F.2 to see an
example of a plan that meets the formatting requirements.

# F    Simulation and Evaluation

## F.1    Simulation Process

### F.1.1    Action Space $\mathcal{A}$

As we have mentioned, there are four kinds of actions permitted to transit to the next state: *transiting*, *resting*, *dining*, and *sightseeing*. In this section, we illustrate the details of how the travel agent takes these actions.

After the traveler completes the event of the previous state, we prompt the traveler to make the decision for the next action as follows:

Based on the plan, determine the next step:
1)To engage in activities at the location (such as visiting POIs, shopping, entertainment, hiking, etc.) .

2) To rest at the location (such as resting at the hotel for a bit, tidying up luggage, or sitting at a resting area within the POI) .

3) To dine at the location, with fast food or snack shop dining times being 1 hour, regular restaurant dining times being 1.5 hours, and high-end restaurant dining times being 2 hours.

4) To proceed immediately to the next destination.

Once the traveler chooses an action, we require the traveler agent to give the output formatted as follows:

***Transiting:*** *{ "decision": "transit", "departure": "...", "destination": "...", "transport mode": "...", "arrival time": "xx:xx", "remaining stamina": ..., "total expense": ..., "next planned location": "..." }*

***Resting:*** *{ "decision": "rest", "end time": "xx:xx", "remaining stamina": ..., "total expense": ..., "next planned location": "..." }*

***Dining:*** *{ "decision": "dine", "end time": "xx:xx", "remaining stamina": ..., "total expense": ..., "next planned location": "..." }*

***Sightseeing:*** *{ "decision": "sightsee", "end time": "xx:xx", "remaining stamina": ..., "total expense": ..., "next planned location": "..." }*

It is noted that only resting, which means resting in place, usually (If you're heading back to the hotel to rest, choose "transit"), does not need additional processing. The following actions, including transiting, dining, and sightseeing, need to be integrated with real-world information.

**Transiting**    If the traveler chooses to transit to the next location according to the plan, there are several modes of transportation, including walking, cycling, public transportation (a combination of walking and bus/metro), and taxis. We use the Amap (Gaode) map API to obtain the travel time and costs of these modes of transportation. The traveler agent needs to comprehensively take into account time, cost, preferences, and stamina to make a choice.

**Dining**    If the traveler chooses to have a meal, we use the Amap (Gaode) map API to find the restaurant near the current location. We provide information on the quality and cost of restaurants for travelers to choose from, which is relevant to the traveler's preferences and budget. Dining usually takes 0.5-2 hours based on the restaurant type, and is also counted in the time of the sandbox environment.

**Sightseeing**    If the traveler chooses to go sightseeing at the current location (usually a POI), we simulate the sightseeing experience of the POI. To be specific, we use the Red Note (Xiaohongshu) API to search for travel blog posts related to POI. Since the blog posts are based on real experiences, we design a sightseeing event agent to simulate the probable experience in advance, taking into account the traveler's preferences, available time, and stamina as follows:

> Today's [Customized Plan] is:
> {day_plan}
>
> Your/your group's initial stamina level is {stamina}, and the current state is "{traveler_state}". The current time is {time}, and you are located at [{location}].
>
> Travel Blog Post Search Results:
> {blog_post_search_results}
>
> [Considerations]
> 1. Estimate the visiting time for each POI and list how you/your group plans to visit each one. Pay attention to the end time, as you need to reserve time for traveling to the next location [{destination_place}].
> 2. Estimate the cost of visiting each POI and list the amount for each expense (such as admission fees, show tickets, etc.).
> 3. According to the [Stamina Rules], estimate the stamina consumption for each visited attraction and describe the stamina consumption situation for each activity during your visit. When your/your group's stamina level is less than 2.0, you need to rest or have a meal to recover stamina and cannot continue visiting. Pay attention to the remaining stamina level and whether it needs to be reserved for the subsequent itinerary.
> 4. Only consider the location [{location}], not other locations after the [Customized Plan]. If it's past the opening hours, you may choose not to visit.
>
> As travelers, based on your current stamina level and time schedule, taking into account the factors mentioned above, how do you plan to proceed with your visit at [{location}]? Please describe how you/your group will conduct the visit.

Although we have simulated an experience of the POI in advance, the traveler can consider whether to actually conduct such an experience, considering the current time, stamina, or other factors.

### F.1.2   Traveler Stamina Engine

**Basic Design**    Many researchers focus on analyzing the preference but neglect the type of traveler. The stamina of different types of travelers is diverse, which significantly influences the number of POIs to be visited in one day. In the travel simulation, we design the rule-based engine to adjust stamina expenditure and recovery, which especially vary for different types of travelers. Besides, each type of traveler possesses an initial stamina value, and the travel group is calculated as a whole.

As shown in Table 8, we showcase some examples from Travel-Sim. For younger people, they have higher initial stamina, consume less energy, and recover faster. For elderly people, they would like to have a more relaxed journey and tend to be exhausted more quickly when sightseeing and transiting.

To enable LLMs to intuitively comprehend what stamina represents, we design a stamina-to-state conversion table that translates stamina values into specific states of the traveler: 1) If stamina is greater than 6.0, the state is "Energetic". 2) If the stamina is greater than or equal to 4.0 and less than 6.0, the state is "Good". 3) If the stamina is greater than or equal to 2.0 and less than 4.0, the state is "Slightly Tired". 4) If the stamina is less than 2.0, the state is "Very Tired".

**Spontaneous Behavior**    Although we do not explicitly instruct the LLM to be aware of the state of the traveler, we find it interesting that the traveler agent can **automatically adjust the next action based on the current state**. For example, if the traveler feels tired, the next action can be "resting in the restaurant" or "transiting by taxi instead of by bus to the next place". In some cases, the traveler modifies the itinerary or skips the next POI depending on the current state, which illustrates that stamina significantly influences the execution of the actual itinerary. This highlights that previous studies, failing to account for the varying stamina of travelers in planning, have resulted in travel plans that are less feasible.

Table 8: Stamina rules for different types of travelers (from examples in Travel-Sim).

| Type | Composition {gender, age} | [Initial stamina] Description | Stamina Rule |
|---|---|---|---|
| Single | {male, 32} | [8.5] An energetic 32-year-old male traveler loves to explore the natural scenery and historical culture, enjoys in-depth experience of local life through public transportation and walking, especially in traditional food and shopping for special souvenirs, preferring to avoid commercial attractions. | sightseeing/1hr: -1; dining: +1; resting/1hr: +1; transiting/1h: bus/metro+0, taxi+0.5, walking-1, cycling-0.5, no cycling |
| Couple | {male, 65}, {female, 62} | [6.5] An elderly couple with a passion for culture and history, preferring a leisurely travel pace. They enjoy visiting temples and savoring local cuisine, all while prioritizing rest and comfort during their journeys. | sightseeing/1hr: -1.5; dining: +0.5; resting/1hr: +0.5; transiting/1h: bus/metro-1, taxi+0, walking-1.5 |
| Family | {male, 32}, {female, 31}, {male, 7}, {female, 4} | [7.0] A family of four consisting of a father who loves natural scenery and technology, a mother who loves shopping and searching for local cuisine, and two active children, they prefer a comfortable and convenient way to travel. The family focuses on parent-child experiences and cultural exploration. | sightseeing/1hr: -1; dining: +0.5; resting/1hr: +0.5; transiting/1h: bus/metro-1, taxi+0, walking-1.5, no cycling |
| Family | {male, 45}, {female, 41}, {female, 71} | [6.5] A family of 45-year-old men, 41-year-old women, and 71-year-old women who prefer relaxing cultural and historical exploration, enjoying local cuisine and special performances, while focusing on the comfort of travel and quality of family time. | sightseeing/1hr: -1.5; dining: +0.5; resting/1hr: +0.5; transiting/1h: bus/metro-1, taxi+0, walking-1.5, no cycling |
| Group | {female, 23}, {female, 24}, {female, 27} | [7.5] A group of three young female travelers who love to explore local food and culture, fashion shopping and photo check-in travel, prefer a relaxed and comfortable way, enjoy lively nightlife, and do not like to wake up early. | sightseeing/1hr: -1; dining: +0.5; resting/1hr: +1; transiting/1h: bus/metro-0.5, taxi+0.5, walking-1, cycling-1 |

## F.2 Case Study

In this section, we showcase an example to intuitively illustrate how the travel simulation works. First of all, we create a traveler agent based on an example in Travel-Sim, e.g., the elderly couple in Table 8. We load the basic information, including the stamina rule and preferences, of the traveler to the system prompt of the LLM agent.

This elderly couple is going to have a 3-day journey in Beijing. They start their journey by arriving at the Beijing West Railway Station at 10:00 AM. Their initial stamina will be subtracted by 2 ($6.5 - 2 = 4.5$) because of the exhaustion of traveling to a new city. The couple can choose one of the four actions to go to the next state. We offer four different routes utilizing various modes of transportation, all provided via the map API:

Day 1 [Custom Itinerary]:
Day 1 – August 11: Arrival in Beijing, Visit to Shichahai

**10:00/[Beijing West Railway Station]:** Arrival at Beijing West Railway Station
• Details for [Beijing West Railway Station]:
  – Transportation Advice: Take a taxi to the Cotton Tree Hotel; approximately a 30-minute drive, convenient and comfortable.
  – Tips: Keep your belongings secure; the station tends to be crowded.

**11:00/[Cotton Tree Hotel]:** Arrive at the hotel and check-in
• Details for [Cotton Tree Hotel]:
  – Check-in Reminder: Confirm check-in time with the hotel in advance. If early check-in isn't available, you can store your luggage.
  – Rest Advice: Take a short rest to prepare for the afternoon tour.

**12:30/[Siji Minfu Restaurant]:** Enjoy a traditional Beijing-style lunch
• Details for [Siji Minfu]:
  – Food Recommendation: Try the Peking Duck; you can request less oil and salt.
  – Dining Tip: Make a reservation in advance to avoid waiting.

**14:30/[Shichahai]:** Explore Beijing's most scenic historical neighborhood
• Details for [Shichahai]:
  – Tour Route: Start at Jinding Bridge → Lotus Market → Yinding Bridge → Yandai Xiejie.
  – Special Activities: Consider a 30-minute boat ride to enjoy the lake view.
  – Rest Suggestion: Take a break at a lakeside tea house and enjoy jasmine tea.
  – Cultural Experience: Walk through the hutongs and immerse yourself in the old Beijing atmosphere.
  – Transport Note: Be prepared for walking; take breaks as needed.

**18:00/[Nanmen Hot Pot]:** Enjoy traditional Beijing-style mutton hot pot
• Details for [Nanmen Hot Pot]:
  – Food Recommendation: Choose traditional copper pot mutton hot pot, adjusting condiments to your taste.
  – Dining Tip: Focus on lean meat to avoid overly greasy options.

**20:00/[Cotton Tree Hotel]:** Return to the hotel for rest
• Details for [Cotton Tree Hotel]:
  – Transportation Advice: Take a taxi back; approximately a 15-minute ride.
  – Rest Reminder: Go to bed early to recharge for the next day's activities.

Your starting stamina is 4.5, and your current state is "Good."
The current time is 10:00, and your location is [Beijing West Railway Station].
So far, you've spent 0 yuan today.

Based on the custom itinerary, do you want to start sightseeing, rest, eat, or move to the next location?

If you wish to proceed to the next destination, here are the available transportation options:
• Walking: [Beijing West Railway Station] → Walk for 122.5 minutes → [Cotton Tree Hotel], cost: 0 yuan
• Public Transport: No public transit available for this route
• Taxi: [Beijing West Railway Station] → Taxi for 37.7 minutes → [Cotton Tree Hotel], cost: 28.0 yuan
• Shared Bike: [Beijing West Railway Station] → Ride a shared bike for 49.0 minutes → [Cotton Tree Hotel], cost: 4.9 yuan

Step 1: As a traveler, reflect on your experience and plan the next steps from a first-person perspective.
Step 2: Analyze stamina consumption (consider the group's overall stamina).
Step 3: Analyze expenses (calculate the total cost based on the number of people).
Step 4: Provide a response in the following JSON format: ...

Considering the travelers' perspective, they decide to take a taxi to their hotel for their next step, as they feel exhausted from the long journey to Beijing:

Step 1: As an elderly couple newly arrived in Beijing, we feel both excited and a bit fatigued. After a long journey, our priority is to get to the hotel as soon as possible, drop off our luggage, and take a short rest. Considering our age, preferences, and the inconvenience of carrying luggage, taking a taxi is the most suitable choice. This will allow us to comfortably reach our destination while conserving energy for the rest of the day's activities. We're looking forward to resting briefly at the hotel, recharging, and preparing for a delightful meal and our next visit to Shichahai.

Step 2: Stamina Consumption Analysis
– Initial stamina: 4.5
– Taking a taxi consumes no stamina
– Final stamina: 4.5 + 0 = 4.5

Step 3: Expense Analysis
– Taxi fare: 28.0 yuan

Step 4: JSON Response
```json
{
    "decision": "transit",
    "departure_location": "Beijing West Railway Station",
    "destination": "Cotton Tree Hotel",
    "transportation_mode": "taxi",
    "arrival_time": "10:38",
    "remaining_stamina": 4.5,
    "total_expenses": 28.0,
    "next_planned_location": "Siji Minfu Restaurant"
}
```

After the couple arrives at the hotel, they choose to rest for a while before heading to the restaurant to taste the local cuisine, e.g., Peking duck:

Step 1: As newly arrived elderly couple in Beijing, we feel both excited and a bit tired. Despite being in good physical condition, considering the fatigue from long-distance travel, we have decided to take some rest at the hotel to prepare for the upcoming schedule. This way, we can better enjoy the local cuisine and attractions in Beijing. We plan to rest at the hotel until around 12:00 PM, then head to Sijiminfu to enjoy a Beijing-style lunch. During our rest, we will organize our luggage and freshen up to make ourselves more comfortable. We are looking forward to tasting Peking duck but will also pay attention to controlling our intake of oil and salt for health reasons.

Step 2: Stamina Consumption Analysis
Initial Stamina: 4.5
Resting at the hotel for approximately 1 hour 20 minutes, energy recovery: 1.33 * 0.5 = 0.665
Final Stamina: 4.5 + 0.665 = 5.165

Step 3: Expense Analysis
No additional expenses so far

Step 4: JSON Response
```json
{
"decision": "rest",
"end_time": "12:00",
"remaining_stamina": 5.165,
"total_expenses": 28.0,
"next_planned_location": "Siji Minfu Restaurant"
}
```

(Skipping the states of transiting and dining) After having a wonderful meal in the restaurant, they plan to visit their first attraction, Shichahai, a historic and scenic area in the heart of Beijing, encompassing three interconnected lakes: Qianhai (Front Sea), Houhai (Rear Sea), and Xihai (West Sea). Known for its well-preserved traditional Siheyuan courtyard houses and Hutongs, Shichahai offers visitors an authentic glimpse into old Beijing culture. To create a simulated sightseeing experience in Shichahai, we utilize the Red Note (Xiaohongshu) API to gather blog posts relevant to visiting this area. Based on the gathered content, we then guide the sightseeing event agent to craft an immersive experience from the perspective of the travelers:

As an elderly couple with a deep love for culture and history, we are very much looking forward to visiting Shichahai in Beijing, a scenic area brimming with ancient charm. Considering our physical condition and preferences, we have decided on the following itinerary:

**2:02 PM – 3:30 PM Qinding Bridge and Lotus Market (1 hour 28 minutes)**
We start from Qinding Bridge, strolling leisurely while enjoying the lake views of Shichahai. Qinding Bridge is an excellent spot for watching the sunset, though it's not evening yet, we can still admire the beautiful lake scenery. Then we walk to Lotus Market to experience the old Beijing's street life.
Cost: Free
Physical exertion: 1.5 * 1.47 ≈ 2.2
Remaining stamina: 5.8225 – 2.2 = 3.6225 (Condition: slightly tired)

**3:30 PM – 4:00 PM Rest at a lakeside teahouse (30 minutes)**
After some time of touring, we feel a bit weary and decide to take a short rest at a teahouse by the lake, savoring a cup of fragrant jasmine tea while enjoying the lake view, recovering our strength.
Cost: Approximately 60 RMB (for two people's tea)
Stamina recovery: 0.5 * 0.5 = 0.25
Remaining energy: 3.6225 + 0.25 = 3.8725 (Condition: slightly tired)

**4:00 PM – 5:00 PM Yinding Bridge and Yandai Xiejie (1 hour)**
After resting, we continue our tour, arriving at Yinding Bridge, another great place to overlook Shichahai. Following that, we meander to Yandai Xiejie, admiring the unique hutong architecture and experiencing the lifestyle of old Beijing.
Cost: Free
Physical exertion: 1.5 * 1 = 1.5
Remaining stamina: 3.8725 – 1.5 = 2.3725 (Condition: slightly tired)

**5:00 PM – 6:00 PM Row boat ride on the lake (1 hour)**
Though our strength has been somewhat depleted, considering that row boating is a characteristic activity of Shichahai, we decide to give it a try. We rent a small boat, gently gliding on the lake, appreciating the natural beauty, and enjoying the leisurely moments. Cost: Approximately 100 RMB (boat rental fee)
Physical exertion: 1.5 * 1 = 1.5
Remaining stamina: 2.3725 – 1.5 = 0.8725 (Condition: very tired)

Summary:
Total visit time: 3 hours 58 minutes
Total cost: 160 RMB
Final stamina status: 0.8725 (very tired)

Our tour largely followed the plan, but due to significant physical exertion, we decide to conclude our visit early and return to the hotel to rest. This arrangement allows us to fully appreciate the beauty and culture of Shichahai while also taking care of our physical condition. Although we did not visit all recommended attractions, we believe this experience was already highly satisfying, leaving us with beautiful memories of our trip to Beijing.

Due to the limited space here, we present only a part of the simulation, but it encompasses most scenarios. This case study showcases that causal travel simulation is based on the integration of real-world information and comprehensively reflects the quality of the generated plan.

```
{
    "type": "couple",
    "number": 2,
    "desc": "This couple in their 30s are travel enthusiasts full of curiosity. They enjoy gaining a deep understanding of their destinations by
trying local cuisine, exploring cultural and historical sites, and participating in unique experiences. They prefer using public transportation
and choose accommodations that are conveniently located and have character.",
    "people": [
        {
            "age": 35,
            "gender": "male"
        },
        {
            "age": 31,
            "gender": "female"
        }
    ],
    "stamina": 7.5,
    "stamina_rule":
    """
    【Stamina Rules】
    The rules for changes in stamina are as follows:
    1. Each hour spent sightseeing deducts 1.0 from stamina.
    2. Each meal can restore 1.0 stamina.
    3. Resting for an hour (such as staying at the hotel or resting at a scenic spot) can restore 1.0 stamina.
    4. Using public transportation for one hour deducts 0.5 from stamina, walking for one hour deducts 1.0 from stamina, taking a taxi
does not deduct stamina, and cycling for one hour deducts 0.5 from stamina.

    Stamina corresponds to the following states:
    1. Stamina greater than or equal to 6.0, the state is "energetic".
    2. Stamina greater than or equal to 4.0 and less than 6.0, the state is "good".
    3. Stamina greater than or equal to 2.0 and less than 4.0, the state is "slightly tired".
    4. Stamina less than 2.0, the state is "very tired".
    """,
    "preferences": {
        "Food Preferences": "They love to try local specialties, including street food and traditional dishes; they are also willing to explore
popular internet celebrity restaurants and highly-rated local eateries.",
        "Shopping Preference": "The woman enjoys shopping, especially for local handicrafts, souvenirs, and designer brand clothing.",
        "Scenery Preference": "Both share a love for natural scenery, such as beaches, mountains, or rural landscapes; they prefer beautiful
spots suitable for photo-taking.",
        "Cultural Interest": "They are enthusiastic about visiting museums, art exhibitions, and other cultural venues to learn about local arts
and culture.",
        "Historical Interest": "They enjoy visiting places with a rich history, such as ancient city ruins and historic buildings, to gain deeper
historical knowledge.",
        "Entertainment Activities": "They enjoy watching live performances like concerts, plays, or dance shows; they also participate in local
festivals or other interesting events.",
        "Sports Activities": "While not particularly interested in sports activities, they might engage in light outdoor activities, such as walking
or cycling.",
        "Leisure Preference": "They tend to keep their itineraries tight, trying to visit as many different places as possible rather than staying
long in one place to relax.",
        "Adventure Preference": "The man has a particular interest in adventurous activities, such as rock climbing, diving, or other thrilling
experiences.",
        "Religious Activities": "They do not have a special interest in religious sites unless they hold significant historical or cultural value.",
        "Learning New Things": "During travels, they hope to learn new things, whether it's language, cooking skills, or crafts.",
        "Transportation Preference": "They prefer public transportation like subways and buses to better integrate into local life and save
costs; they may occasionally take taxis to save time; they enjoy walking to explore alleys and streets; under special circumstances, they
might rent a car for self-drive tours.",
        "Accommodation Preference": "They tend to choose mid-range hotels or distinctive bed and breakfasts that are conveniently
located, clean and comfortable, preferably close to major tourist attractions or transportation hubs; they value safety and service quality;
they don't have high requirements for room facilities; if possible, they will choose accommodations with romantic elements, such as sea-
view rooms or thematically decorated rooms.",
        "Other Preferences": "They seek unique travel experiences, such as hot air balloon rides over the city or night boat tours; they are
happy to start a new day early and usually choose public transportation to experience the local atmosphere."
    },
    "Total Accommodation Budget": "400-800 RMB per night",
    "Total Sightseeing Budget": "600-1000 RMB per day"
}
```

Figure 9: An example in Travel-Sim. The traveler group consists of a couple with one female and one male in their 30s. They have an initial stamina of 7.5 and consume less stamina for walking and cycling compared to elders and families with kids. They have specific and detailed preferences that will greatly influence their decision on the itinerary.

## F.3 Travel-Sim Dataset Card

As shown in the example in Figure 9, we construct the Travel-Sim dataset that consists of diverse travelers with detailed preferences. Different from previous travel planning benchmarks that only consider single travelers with limited preferences, Travel-Sim accommodates various types of travelers with diverse group compositions, such as individuals, couples, groups, families, and more. Each traveler type may have distinct preferences regarding activities, accommodation standards, budget constraints, travel pace, and cultural experiences.

To build such a dataset, we annotate the profiles of 16 types of travelers and leverage Deepseek-R1 to expand the details. We select the 7 most-visited cities in China as the destinations and combine them with the travelers to generate 112 {traveler, destination} pairs as the evaluation dataset.

Table 9: Datasets with City Composition.

| Dataset | City List | City Count | Traveler Type Count |
|---------|-----------|------------|---------------------|
| **Reward Model Training** | Guilin, Qingdao, Luoyang, Xishuangbanna, Shenyang, Wuhan, Nanchang, Zhengzhou, Changchun, Xianyang, Lanzhou, Yangzhou, Chaozhou, Guangzhou | 14 | 37 |
| **RFT/RL Training** | Beijing, Chongqing, Nanjing, Chengdu, Datong, Jingdezhen, Dalian, Hangzhou, Beihai, Lijiang, Kunming, Taiyuan, Xianning, Shenzhen, Hong Kong, Changsha, Shantou, Qinhuangdao, Enshi, Tianjin | 20 | 120 |
| **Evaluation** | Lhasa, Sanya, Shanghai, Harbin, Dali, Xi'an, Xiamen | 7 | 16 |

## F.4 Multi-granularity Evaluation by Traveler

After experiencing the itinerary, the traveler has much to share regarding it. We implement a multi-granularity evaluation mechanism for the travel experience from three levels: the traveler reflects on the experience and rates the score after each POI visit, at the day's conclusion, and after completing the entire journey. We first ask the traveler agent to think from the perspective of a traveler by outputting the psychological activities. We then ask the traveler agent to rate the score by inspecting five indicators, e.g., travel experience (ex), scenic spot characteristics (it), sightseeing arrangements (ar), stamina exertion (st), and overall expense (co).

**For every end of the visit to the POI**, we let the traveler assess the POI visiting experience as follows:

```
[Evaluation Criteria]
The evaluation criteria include: travel experience, scenic spot characteristics, sightseeing arrangements, stamina exertion from
sightseeing, and overall expense.
1. Travel Experience: How was the travel experience? Did you feel that the activities in the attraction were fulfilling? Were they in line with
your preferences? Are there any regrets?
2. Scenic Spot Characteristics: What special features do the scenic spots have? Were there any special activities or views? Are these
places of interest to you?
3. Sightseeing Arrangements: Was there enough time to fully enjoy each attraction? Were the timing and conditions ideal (too
crowded/too few people, missed activities or best viewing times)? Were there any locations not visited due to lack of time or closure?
4. Stamina Exertion from Sightseeing: Was the schedule too tight or too relaxed? Was it overly exhausting? Was there sufficient time for
meals and rest (were breakfast, lunch, and dinner arranged)?
5. Overall Expense: Was today's overall spending within budget? Were there areas of overspending? Were there opportunities to save
money?

Please rate your visit to [{location}] on a scale of 1 to 5.
First, analyze these five indicators based on the traveler's psychological activities, then return the following JSON format:
```json
{
    "Decision": "Evaluation",
    "Travel Experience Rating": ...,
    "Scenic Spot Characteristics Rating": ...,
    "Sightseeing Arrangements Rating": ...,
    "Stamina Exertion from Sightseeing Rating": ...,
    "Overall Expense Rating": ...
}
```

**For every end of the whole-day itinerary**, we let the traveler assess the travel experience of today as follows:

Today's itinerary has come to an end. What do you think about the arrangement of this [customized plan]? Please evaluate today's activities.

[Evaluation Criteria]
The evaluation criteria include: enjoyment experience, scenic spot characteristics, sightseeing arrangements, and stamina exertion from sightseeing, and overall expense.
1. Travel Experience: How was the experience? Was the day's activities fulfilling? Did it align with your preferences? Were there any regrets?
2. Scenic Spot Characteristics: What special features did the scenic spots have? Were there any special activities or views? Were these places of interest to you?
3. Sightseeing Arrangements: Was there enough time to fully enjoy each attraction? Were the timing appropriate (too crowded/too few people, missed activities or best viewing times)? Were there any locations not visited due to lack of time or closure?
4. Staminia Exertion from Sightseeing: Was the schedule too tight or too relaxed? Was it overly exhausting? Was there time for meals and rest (were breakfast, lunch, and dinner arranged)?
5. Overall Expense: Was today's overall spending in line with the budget? Were there any overspending areas? Were there opportunities to save money?

Please rate your sightseeing experience today on a scale of 1 to 5.
First, analyze these five indicators based on the traveler's psychological activities, then return the following JSON format:
```json
{
    "Decision": "Evaluation",
    "Travel Experience Rating": ...,
    "Scenic Spot Characteristics Rating": ...,
    "Sightseeing Arrangements Rating": ...,
    "Staminia Exertion from Sightseeing Rating": ...,
    "Overall Expense Rating": ...
}
```

**At the end of the multi-day journey**, we let the traveler assess the overall travel experience of the entire trip as follows:

Now, you can evaluate the entire [customized plan] based on your travel experience.

[Evaluation Criteria]
The evaluation criteria include: travel experience, scenic spot characteristics, sightseeing arrangements, stamina exertion from sightseeing.
1. Travel Experience: How was the travel experience? Did you feel that the day's activities were fulfilling? Were they in line with your preferences? Are there any regrets?
2. Scenic Spot Characteristics: What special features do the scenic spots have? Were there any special activities or views? Are these places of interest to you?
3. Sightseeing Arrangements: Was there enough time to fully enjoy each attraction? Were the timing and conditions ideal (too crowded/too few people, missed activities or best viewing times)? Were there any locations not visited due to lack of time or closure?
4. Stamina Exertion from Sightseeing: Was the schedule too tight or too relaxed? Was it overly exhausting? Was there sufficient time for meals and rest (were breakfast, lunch, and dinner arranged)?
5. Overall Expense: Was today's overall spending within budget? Were there areas of overspending? Were there opportunities to save money?

Based on your evaluation, please rate your {len(self.everyday_plan)} days of travel experiences, with a full score of 5 points.
First, analyze these five indicators based on the traveler's psychological activities, then return the following JSON format:
```json
{
    "Decision": "Evaluation",
    "Travel Experience Rating": ...,
    "Scenic Spot Characteristics Rating": ...,
    "Sightseeing Arrangements Rating": ...,
    "Stamina Exertion from Sightseeing Rating": ...,
    "Overall Expense Rating": ...
}
```

### F.5 Human Verification for Simulation-based Evaluation

Although the simulated travel empirically seems to be effective for evaluating the generated plan, we further implement human evaluation to verify if the simulation-based evaluation is consistent with the human evaluation. To be specific, we provide the evaluation results from the simulated travel for humans to check if the evaluation from the traveler agent is reasonable. For example, given the evaluation of the POI travel experience, humans inspect it based on the map and blog post, subsequently deciding whether to endorse the evaluation. If humans agree with the travel agent's evaluation, the scores remain the same as those of the travel agent. If humans disagree, they have the option to modify the scores.

Three individuals take part in this experiment. We record the modified scores and the deviation values to calculate the agreement rates. It is noted that the scores here are PER-ex, PER-it, PER-ar, PER-st, and PER-co.

As shown in Table 10, the evaluation of traveler agents has high consistency with the human evaluation, with the agreement rate of 92%. We have identified that the PER-co (travel cost) metric exhibits a relatively higher level of inconsistency. The travel agent often buys too many expensive souvenirs,

which leads to going over budget. Additionally, they occasionally make mistakes when calculating total travel expenses, especially when accounting for a group of travelers.

Table 10: We implement human evaluation to verify if the simulation-based evaluation is consistent with humans'.

|  | ex | it | ar | st | co | PER-agg. | Agree. rate |
|---|---|---|---|---|---|---|---|
| R1-Distill 7B (s.) + R1-Distill 7B (p.) | 77.5 | 86.4 | 79.3 | 76.4 | 87.6 | 81.4 | - |
| Human | 69.2 | 83.7 | 73.4 | 74.2 | 74.5 | 75.0 | 92.1% |

# G   Metrics

We examine four criteria to evaluate the effectiveness of wide-horizon thinking. Comprehensiveness (CPH) and Completeness (CPL) are rule-based metrics, while Feasibility (FEA) and Personalization (PER) are simulation-based metrics.

## G.1   Rule-based Metrics

### G.1.1   Comprehensiveness (CPH)

Comprehensiveness (CPH) evaluates how much relevant information is effectively integrated from the long context into the final plan. To elaborate, for each POI in the plan, we first extract the corresponding travel guidance. We calculate the similarity between the POI travel guidance and POI-related blog posts in the context. We encode both texts into embedding and calculate the cosine similarity [3]. We calculate the average similarity across all POIs in the plan as the comprehensive score.

### G.1.2   Completeness (CPL)

To create an organized travel plan, we must include several essential elements, such as the timeline, destinations, activities, etc. We require LLMs to generate a travel plan that follows the specific output structure and includes the required elements. To evaluate if the generated plan strictly follows formatting instructions for a thorough itinerary, we inspect four criteria as follows:

1. The origin and destination of the entire journey must be at the specified stations or airports.

2. Each day's itinerary, excluding the first and last day, will begin and end at the hotel where the traveler is accommodated.

3. The introduction for each point of interest's sightseeing should be formally structured according to the format specified in the prompt.

4. In the travel itinerary, activities should be scheduled to include meal arrangements for both lunch and dinner.

For criteria 1, 2, and 3, we use the regex expression to extract the keywords and verify whether the generated plan satisfies these criteria. For the last one, we leverage the Deepseek-R1 [18] to verify if the itinerary of each day includes arrangements for lunch and dinner. We evaluate each plan against these four criteria, awarding 25 points for each criterion that is met. Therefore, for each plan, we have a maximum of 100.0 points for CPL by summing all the scores.

## G.2   Simulation-based Metrics

Because some criteria, e.g., feasibility and personalization, are hard and less persuasive to be simply measured by rule-based metrics, we deal with them via travel simulation based on the real world.

---

[3]We use the *paraphrase-multilingual-mpnet-base-v2* model of the *sentence-transformers* library in `https://sbert.net/docs/sentence_transformer/pretrained_models.html`.

### G.2.1 Feasibility (FEA): Travel Plan Similarity Score

We evaluate the feasibility of the generated plan by ensuring that the plan is realistic and executable within the given constraints, such as time and spatial optimization. As shown in Figure 5, we can perceive the generated travel plan as a trajectory of multiple "time-location" pairs. As the traveler conducts a virtual journey based on the generated travel plan, the traveler also produces a trajectory of "time-location" pairs. We aim to assess the feasibility of the generated plan by evaluating whether the plan aligns with the traveler's itinerary.

To calculate the similarity between two trajectories of "time-location" pairs, there are a few things to consider:

1. **Completeness**: The events of two trajectories should match. Missing or extra events should be penalized.

2. **Chronological order**: The events of two trajectories should happen in the same order.

3. **Temporal proximity**: The identical events of these two trajectories should occur as closely in time as possible. A penalty shall be imposed that corresponds to the temporal discrepancy between identical events. The magnitude of this penalty increases proportionally with the extent of the time gap.

Based on the considerations above, as shown in Algorithm 3, we design an algorithm for calculating the similarity between two trajectories of "time-location" pairs, dubbed **Travel Plan Similarity Score (TPSS)**. We separately deal with the similarity of time and locations and implement dynamic programming to iteratively calculate the score. It is noted that we use TPSS to calculate the similarity of trajectories of one day. For multi-day journeys, we calculate the final TPSS by averaging the daily scores.

---

**Algorithm 3** Travel Plan Similarity Score Calculation

---

**Input:** generated_plan_trajectory $T_g$, simulated_travel_trajectory $T_s$
**Output:** similarity score (as percentage)
**function** CALCULATEPLANSIMILARITY($T_g$, $T_s$)
    $m \leftarrow$ length of $T_g$
    $n \leftarrow$ length of $T_s$
    Initialize $dp[m+1][n+1]$ with zeros
    **for** $i \leftarrow 1$ to $m$ **do**
        **for** $j \leftarrow 1$ to $n$ **do**
            $score \leftarrow$ CalculateMatchScore($T_g$[i-1], $T_s$[j-1])
            $dp[i][j] \leftarrow \max(dp[i-1][j], dp[i][j-1], dp[i-1][j-1] + score)$
        **end for**
    **end for**
    $max\_score \leftarrow \min(m, n)$                     ▷ Ideal maximum score
    $similarity \leftarrow dp[m][n]/max\_score$
    $completeness\_penalty \leftarrow \min(m, n)/\max(m, n)$    ▷ Penalize for missing/extra activities
    $final\_similarity \leftarrow similarity * completeness\_penalty * 100$
    **return** $final\_similarity$
**end function**
**function** CALCULATEMATCHSCORE(item1, item2)
    $time\_score \leftarrow$ TimeDiffScore(item1.time, item2.time)
    **if** item1.location = item2.location **then**
        $location\_score \leftarrow 1$
    **else**
        $location\_score \leftarrow 0$
    **end if**
    **return** $(time\_score + location\_score)/2$
**end function**
**function** TIMEDIFFSCORE(time1, time2)
    $diff \leftarrow$ |time2 - time1| in hours
    **return** $\max(0, 1 - diff/2)$                   ▷ Linear decrease within 2 hours
**end function**

---

### G.2.2 Personalization (PER): Aggregated Assessment from Travelers

We evaluate the personalization of the generated plan by inspecting whether the travel plan meets the unique needs and preferences of the traveler agent in the simulation. As we have mentioned, our multi-granularity evaluation mechanism via traveler's assessment in Appendix F.4, we further introduce how we aggregate the scores from multi-granularity assessment in Equation 3 as follows:

$$\mathcal{S}_{PER} = \alpha_1 \mathcal{S}_{whole\_travel} + \alpha_2 \frac{1}{N} \sum_i^N (\beta \mathcal{S}_{day\_i} + \gamma \mathcal{S}_{POI\_day\_i}). \tag{3}$$

As each POI has a score after sightseeing, we average the scores of each POI to be the overall POI score $\mathcal{S}_{POI\_day\_i}$ in the $i^{th}$ day. We weight the $\mathcal{S}_{POI\_day\_i}$ and the score of the whole-day itinerary $\mathcal{S}_{day\_i}$ by $\beta$ and $\gamma$ as the aggregated score of $i^{th}$ day. We set $\beta$ as 0.6 and $\gamma$ as 0.4. We average the aggregated score of each day and weight it with the score of the whole travel by $\alpha_1$ and $\alpha_2$. We set $\alpha_1$ as 0.6 and $\alpha_2$ as 0.4. We normalize the $\mathcal{S}_{whole\_travel}$ to 100.0 as the final score.

