# OpenReview forum: "Wide-Horizon Thinking and Simulation-Based Evaluation for Real-World LLM Planning with Multifaceted Constraints"
_NeurIPS.cc/2025/Conference — NeurIPS 2025 spotlight_

### Official Review · Reviewer_uXfj · 2025-06-17

**Clarity:** 2
**Significance:** 4
**Originality:** 3
**Rating:** 4
**Confidence:** 3

**Summary:**

The authors formulated travel planning as an $L^3$ planning problem (long context, long instruction, and long output). This type of wide-horizon thinking is characterized by parallel information processing followed by information integration at each decision step. To handle wide-horizon thinking, the authors proposed Multiple Aspects of Planning (MAoP). In this approach, a strategist first decomposes the planning request into independent aspects, then integrates these aspects into a coherent planning blueprint.
The MAoP training consists of three components: reward model training, rejection sampling fine-tuning for the strategist model, and reinforcement learning training for the planning model. MAoP inference includes pre-planning (decomposition and routing) and aspect-aware thinking, which allows the planning blueprint to be turned into the final plan.
To train their agent, the authors constructed the Travel-Sim benchmark. To evaluate the experimental results, the authors proposed several metrics. For information integration, they used comprehensiveness (CPH), completeness (CPL), feasibility (FEA), and personalization (PER). For user experience, they assessed experience (ex), interest (it), arrangement (ar), stamina (st), and cost (co). The experiments verified the effectiveness of MAoP.

**Questions:**

Q1. You proposed MAoP to overcome the limitations of wide-horizon thinking. However, you also mentioned that MAoP is designed for the proposed $L^3$ planning problem, but this concept is barely discussed in the methods and experiments sections. Do you think removing the L3 planning problem concept could improve the readability of your paper?

Q2. You mentioned that the agent is trained in Travel-Sim. Did you perform any fine-tuning using real-world data? How do you view the sim2real gap, and what are your plans to overcome it?

Q3. Do you have any plan to open-source your code?

**Ethical Concerns:**

["NO or VERY MINOR ethics concerns only"]

**Limitations:**

yes

**Quality:**

2

**Strengths And Weaknesses:**

Strengths:

1. The proposed MAoP demonstrates a strong real-world application for travel planning, and the travel planning pipeline is practical.
2. The Travel-Sim benchmark would be valuable to the community if the authors could make it publicly available.

Weaknesses:

1. The proposal of unnecessary concepts affects readability.
2. The code is not available.

---

> ### Author Rebuttal · Authors · 2025-07-30
>
> We are very grateful for your thorough review and highly insightful comments. Your suggestions were instrumental in helping us strengthen our arguments and improve the paper's clarity.
>
> **Q1:** You proposed MAoP to overcome the limitations of wide-horizon thinking. However, you also mentioned that MAoP is designed for the proposed $L^3$ planning problem, but this concept is barely discussed in the methods and experiments sections. Do you think removing the $L^3$ planning problem concept could improve the readability of your paper?
>
> **A1:** We introduce the $L^3$ planning problem to formalize the specific planning tasks like travel planning, which necessitate wide-horizon thinking. Its fundamental challenge is the need to integrate a wide array of parallel and often conflicting information sources and constraints. This problem is typically characterized by a long context (presentation of the information and constraints), long instructions (how to deal with the information and constraints), and a long output (the final plan). The core challenge MAoP addresses is not specific to travel but to different $L^3$ planning problems, stemming from its Strategist-Planner framework to manage this "wide-horizon" complexity.
>
> Here is another example of $L^3$ planning problem that MAoP can potentially be generalized to:
>
> **Complex Project Management (e.g., a product launch):**
> *   **$L^3$ Challenge:** A project manager must synthesize market research, budget constraints, team capabilities, competitor analysis, and stakeholder requirements into a single, coherent launch plan.
> *   **MAoP Application:**
>     * The `Strategist` would decompose the task into aspects like Budget Allocation, Marketing Channel Strategy, Content Creation, Risk Assessment, and KPIs.
>     * Its `Routing` function would create a blueprint, ensuring that budget decisions inform channel selection, which in turn dictates content needs.
>     * The `Planner` would then use this blueprint to generate a comprehensive project plan, holistically balancing all factors.
>
>
> According to your concerns, **we will revise the manuscript to better integrate this concept (or rename it) and clarify its connection to our methods and experiments, improving readability.**
>
>
> **Q2:** You mentioned that the agent is trained in Travel-Sim. Did you perform any fine-tuning using real-world data? How do you view the sim2real gap, and what are your plans to overcome it?
>
> **A2:** The Travel-Sim dataset (including the reward model training, RFT/RL training and evaluation datasets) comprises three key components: destination city data, preprocessed information, and traveler profiles.
>
> **1.Destination city data**
>
> We select cities from the list of China's Excellent Tourism Cities (a city title issued by the National Tourism Administration of China). For each dataset, we ensure selected cities have no overlap while comprehensively covering diverse geographical, cultural, and environmental characteristics.
>
> | Dataset| City List | City Count | Traveler Type Count |
> |--------------|-----------|------------|-------------------|
> | **Reward Model Training** | 桂林(Guilin), 青岛(Qingdao), 洛阳(Luoyang),  西双版纳(Xishuangbanna), 沈阳(Shenyang), 武汉(Wuhan), 南昌(Nanchang), 郑州(Zhengzhou), 长春(Changchun), 咸阳(Xianyang), 兰州(Lanzhou), 扬州(Yangzhou), 潮州(Chaozhou), 广州(Guangzhou) | 14 | 37 |
> | **RFT/RL Training** | 北京(Beijing), 重庆(Chongqing), 南京(Nanjing),  成都(Chengdu), 大同(Datong), 景德镇(Jingdezhen), 大连(Dalian), 杭州(Hangzhou), 北海(Beihai), 丽江(Lijiang), 昆明(Kunming), 太原(Taiyuan), 咸宁(Xianning), 深圳(Shenzhen), 香港(Hong Kong), 长沙(Changsha), 汕头(Shantou), 秦皇岛(Qinhuangdao), 恩施(Enshi), 天津(Tianjin) | 20 | 120 |
> | **Evaluation** | 拉萨(Lhasa), 三亚(Sanya), 上海(Shanghai), 哈尔滨(Harbin),  大理(Dali), 西安(Xi'an), 厦门(Xiamen)| 7 | 16 |
>
> **2. Preprocessed information**
>
> As illustrated in Appendix C's preprocessing workflow, we gather real-world data through multiple channels, including:
>
> - Interactive agent simulations
>
> - Travel blog content analysis
>
> - Search engine data extraction
>
> - Map API integrations
>
> For instance, attraction information is systematically processed from both travel blogs and search engine results, while geographical data is collected through map APIs.
>
> **3. Traveler profiles**
>
> For traveler profiles, we begin by **manually** creating 30 distinct traveler archetypes based on real-world travel advice requests from blog posts. These initial profiles are then refined using Gemini-2.0-Pro-Exp-0205 to serve as a high-quality seed dataset. To expand the dataset, we employ Gemini-2.0-Pro-Exp-0205 for synthetic data generation. All synthesized profiles undergo **manual human review and filtering** to guarantee accuracy and diversity.
>
> **Therefore, most of the data in the finetuning datasets is from real-world data (except for the traveler types). We try to minimize the sim2real gap to leverage as much real-world data as possible, including real travel blog posts, search engine results, and map APIs.**
>
>
>
> **Q3:** Do you have any plan to open-source your code?
>
> **A3:** We will release the code and open-source the dataset shortly, following an internal inspection of the dataset (as some of the travel blog posts within it are obtained via an internal corporate API). **Anyone who adheres to the license in Appendix B may use our code and dataset.**

---

### Official Review · Reviewer_RxvN · 2025-07-01

**Clarity:** 3
**Significance:** 3
**Originality:** 2
**Rating:** 4
**Confidence:** 4

**Summary:**

The paper tackles the challenge of producing high‑quality travel itineraries with large language models (LLMs) when the task involves long context (many heterogeneous facts), long instructions (formatting rules, chain‑of‑thought prompts), and long outputs (multi‑day plans). The authors name this the L³ planning problem. They observe that conventional “long‑horizon” reasoning (deep, step‑by‑step chains) is ill‑suited to travel planning, which instead demands wide‑horizon thinking—parallel consideration of many distinct aspects such as budget, stamina, spatial constraints, and personal preferences. The paper propose MAoP (Multiple Aspects of Planning), use strategist model performing pre‑planning. Strategist can decompose the user request into aspect‑specific guidance snippets and then “routes” or aggregates these aspects into a coherent planning blueprint that captures cross‑aspect interactions. They also propose Travel‑Sim which can evaluate their method more better. Overall, the work contributes a novel planning framework (MAoP), a distillation strategy for small models, and an agent‑based evaluation benchmark (Travel‑Sim) that together advance LLM‑based solutions for complex, real‑world planning tasks.

**Questions:**

1) Have you considered validating Travel-Sim’s user satisfaction results with real human feedback to confirm its alignment with human perception?

2) Can you provide more discussion or experiments on how MAoP might generalize to wide-horizon planning problems outside of travel?

3) What specific innovations differentiate your Strategist-planner framework from prior modular or hierarchical planning approaches?

**Ethical Concerns:**

["NO or VERY MINOR ethics concerns only"]

**Final Justification:**

Thank you for the further explanations. I appreciate the authors’ detailed clarifications regarding my concern. However, I still feel that the generalizability of MAoP would be more convincing if the paper included concrete experiments beyond the travel planning domain. While the mentioned applications in project management and financial advising are reasonable, without empirical demonstrations in these areas, the broader impact remains more speculative than validated.

Nonetheless, I agree that the paper tackles an interesting and practical problem, and the proposed wide-horizon planning framework represents a novel and useful direction for LLM-based planning systems. Despite the somewhat limited evaluation scope, I believe the paper offers meaningful contributions. I will therefore maintain my original positive rating.

**Limitations:**

yes

**Quality:**

3

**Strengths And Weaknesses:**

Strengths:

1) This paper addresses a real and growing use case: Travel planning, which is a realistic, high-impact application for LLMs.

2) The authors go beyond standard rule-based metrics by proposing Travel-Sim, a realistic agent-based evaluation framework with multi-granularity, multi-dimension feedback. The combination of objective trajectory similarity and subjective traveler satisfaction is a notable improvement over prior evaluation strategies.

2) The paper presents quantitative improvements over several baselines, and shows how MAoP scales better with the number of planning aspects. The experimental protocol is clearly described, with relevant baselines and ablations.


Weaknesses:

1) While Travel-Sim is creative, it is still an LLM simulation of user satisfaction, which may not fully reflect human perception.

2) Though tourism is a strong testbed, the paper could further discuss how MAoP generalizes to other wide-horizon planning settings beyond travel. Without this, its broader impact may be undercut.

3) Strategist-planner separation is a logical extension of existing modular or hierarchical planning ideas. While well-executed, Strategist is  adaptation of existing techniques rather than fundamentally new.

---

> ### Author Rebuttal · Authors · 2025-07-30
>
> We would like to thank you for your constructive feedback and insightful suggestions.
>
> **W1&Q1:** While Travel-Sim is creative, it is still an LLM simulation of user satisfaction, which may not fully reflect human perception.
>
> **A1:**
> **We include the human evaluation to verify the alignment of simulation-based evaluation with human perception, providing experimental details in Appendix F.5.** As shown in the following table of PER score evaluation for "R1-Distill 7B (s.) + R1-Distill 7B (p.)", the evaluation of traveler agents has **high consistency with the human evaluation, with the agreement rate of 92% (PER-agg. ratio)**. We also identify that the PER-co (travel cost) metric exhibits a relatively higher level of inconsistency. We find that the traveler agents sometimes buy souvenirs but do not include them in the travel budget. In these cases, human evaluators give lower PER-co scores due to the budget overruns.
>
> |  | ex | it| ar | st | co | PER-agg. | Agree. rate|
> |------|---------------|----------|-------------------|--------------|---------|---------|---------|
> Simulation| 77.5 | 86.4 | 79.3 | 76.4 | 87.6 | 81.4 | - |
> Human | 69.2 | 83.7 | 73.4 | 74.2 | 74.5 | 75.0 | 92.1%|
>
>
> **W2&Q2:** Though tourism is a strong testbed, the paper could further discuss how MAoP generalizes to other wide-horizon planning settings beyond travel. Without this, its broader impact may be undercut.
>
> **A2:** We appreciate this crucial point. While tourism serves as a intuitive testbed, the architectural principles of MAoP are intentionally designed to be domain-agnostic. **The core challenge MAoP addresses is not specific to travel but to any task where the primary difficulty is integrating a wide array of parallel, often conflicting, information sources and constraints, rather than executing a deep sequence of actions [1].** The framework's generalizability stems from its unique approach to managing this "wide-horizon" complexity.
>
> Here are a few examples of how MAoP can be generalized:
>
> 1.  **Complex Project Management (e.g., a product launch):**
>     *   **Wide-Horizon Challenge:** A project manager must synthesize market research, budget constraints, team capabilities, competitor analysis, and stakeholder requirements into a single, coherent launch plan.
>     *   **MAoP Application:**
>         * The `Strategist` would decompose the task into aspects like Budget Allocation, Marketing Channel Strategy, Content Creation, Risk Assessment, and KPIs.
>         * Its `Routing` function would create a blueprint, ensuring that budget decisions inform channel selection, which in turn dictates content needs.
>         * The `Planner` would then use this blueprint to generate a comprehensive project plan, holistically balancing all factors.
>
> 2.  **Personalized Financial Advising:**
>     *   **Wide-Horizon Challenge:** An advisor must create a portfolio by integrating a client's risk tolerance, age, income, long-term goals (retirement), short-term needs (liquidity), and complex market data with tax implications.
>     *   **MAoP Application:**
>         * The `Strategist` would identify aspects such as Risk Profile, Asset Allocation, Tax Optimization, and Liquidity Planning.
>         * It would `route` these aspects to ensure the final portfolio is a logical synthesis of the client's profile and goals.
>         * The `Planner` would generate the detailed investment strategy document.
>
> In each case, MAoP's core innovations—decomposing the *problem space* into parallel aspects rather than the *solution process* into sequential steps, and using a strategist to intelligently "route" these aspects, providing a scalable method for complex synthesis.
>
>
> **W3&Q3:** Strategist-planner separation is a logical extension of existing modular or hierarchical planning ideas. While well-executed, Strategist is an adaptation of existing techniques rather than fundamentally new.
>
> **A3:** While our Strategist-Planner (MAoP) framework builds on the established principle of modularity, it introduces fundamental innovations that distinguish it from prior art.
>
> First, it shifts the planning philosophy from *long-horizon thinking*—managing a deep sequence of actions, as seen in frameworks like Plan-and-Act [1]—to *wide-horizon thinking*. Our approach is designed for $L^3$ problems where the challenge is synthesizing a *breadth* of parallel constraints, not the *length* of an action sequence.
>
> This leads to a different decomposition target. Unlike hierarchical planners such as ReAcTree [2], which break down the solution process into sequential sub-goals, our strategist decomposes the problem space into concurrent "aspects" (e.g., budget, user preferences, transportation).
>
> Finally, the strategist's two-stage *"Decompose-then-Route"* process is a novel mechanism, not an adaptation. The routing step explicitly filters and structures these aspects into a coherent blueprint to overcome the scalability limits of planning, which is a problem not addressed by traditional planners [3]. This prevents performance degradation from information and constraints overload.
>
> [1] Erdogan, Lutfi Eren, et al. "Plan-and-act: Improving planning of agents for long-horizon tasks." arXiv preprint arXiv:2503.09572 (2025).
>
> [2] Choi, Jae-Woo, et al. "Reactree: Hierarchical task planning with dynamic tree expansion using llm agent nodes." (2025).
>
> [3] Xie, Jian, et al. "Travelplanner: A benchmark for real-world planning with language agents." arXiv preprint arXiv:2402.01622 (2024).

---

> > ### Comment · Reviewer_RxvN · 2025-08-04
> >
> > Thank you for the further explanations. I appreciate the authors’ detailed clarifications regarding my concern. However, I still feel that the generalizability of MAoP would be more convincing if the paper included concrete experiments beyond the travel planning domain. While the mentioned applications in project management and financial advising are reasonable, without empirical demonstrations in these areas, the broader impact remains more speculative than validated.
> >
> > Nonetheless, I agree that the paper tackles an interesting and practical problem, and the proposed wide-horizon planning framework represents a novel and useful direction for LLM-based planning systems. Despite the somewhat limited evaluation scope, I believe the paper offers meaningful contributions. I will therefore maintain my original positive rating.

---

> > > ### Author Response · Authors · 2025-08-05
> > >
> > > Thank you for your positive rating and constructive feedback. We strongly agree with your point regarding generalizability; it is indeed a key direction for deepening this research. In our future work, we will actively apply MAoP to more task domains to validate its broader effectiveness. Thank you again for your affirmation and support.

---

### Official Review · Reviewer_Jw3b · 2025-07-01

**Clarity:** 1
**Significance:** 3
**Originality:** 4
**Rating:** 4
**Confidence:** 2

**Summary:**

This paper introduces a comprehensive approach for travel planning utilizing large language models (LLMs), implemented through the development of two key components: a Multi-aspect Planning (MAoP) framework designed to enhance LLM planning capabilities by addressing three critical challenges—long context, long instruction, and long output—and an agent-based evaluation system called Travel-Sim that overcomes fundamental limitations in evaluation methodologies for complex real-world scenarios through the simulation of real-world travel experiences.

**Questions:**

Questions and Concerns:
1. What is the purpose of One-Step Wide-Horizon Thinking in Section 3.3? Experimental evidence is needed to support the rationale for adopting a one-step approach.
2. What is the detailed formulation of the reward function used in the RFT and RL processes?
3. Could the preprocessing framework workflow be included in the main text to facilitate reader understanding?
4. What is the source of the synthetic data used for training the reward model in MAoP?
5. What is the purpose of the comparison in Section 5.2.4? How does this relate to your proposed Consistent Travel Simulation?
6. Could you include an evaluation of the simulation itself? The current experiments do not demonstrate the reliability of the simulation component.

**Ethical Concerns:**

["NO or VERY MINOR ethics concerns only"]

**Final Justification:**

After carefully reviewing the authors' responses and all reviewers' discussions, I confirm they have fully addressed my concerns and provided additional experimental results. As such, I will revise my rating accordingly. The paper now meets the conference's standards for publication.

**Limitations:**

The authors discussed some limitations and broader impacts.

**Quality:**

2

**Strengths And Weaknesses:**

Strengths:
The MAoP framework addresses broad-scope reasoning challenges through a strategy-planning model architecture that decomposes complex planning into manageable, aspect-oriented components, making it particularly well-suited for travel planning tasks.
The paper proposes a comprehensive evaluation framework incorporating innovative metrics that provides solid support for experimental validation.
The successful distillation to smaller models and integration with real-world data sources (maps, travel blogs) demonstrates immediate applicability for consumer-facing travel planning services.

Weaknesses:
The methodology section suffers from confusing logic and organization, making it easy for readers to become misled or confused.
The experimental section lacks sufficient description of the datasets, providing only macro-level descriptions without in-depth elaboration.
The paper lacks comparative experiments and fails to compare against other existing methods, presenting only comparisons between different large language models.

---

> ### Author Rebuttal · Authors · 2025-07-30
>
> Thank you for your time and valuable comments. We have carefully considered your suggestions, which have been a great help in improving our manuscript.
>
> **W1:** The method section suffers from confusing logic and organization, making it easy for readers to be confused.
>
> **A1:**
>  We wish to clarify our methodology, which is organized into two primary, logical contributions: a novel planning method (MAoP) and a novel evaluation benchmark (Travel-Sim).
>
> **1. The MAoP Method (Plan Your Travel):**
>
> → **Definition**: We first define a specific complex planning task as the $L^3$ planning problem. Its fundamental challenge is the need to integrate a wide array of parallel and often conflicting information sources and constraints. This problem is typically characterized by a long context (presentation of the information and constraints), long instructions (how to deal with the information and constraints), and a long output (the final plan).
>
> → **Preliminaries**: We use the travel planning as the testbed. We find simple "wide-horizon thinking" approach outperforms standard methods but still fail to connect different aspects and scale effectively.
>
> → **Solution**: MAoP overcomes this by using a two-model system (Strategist-Planner). A strategist performs high-level "pre-planning", decomposing the request into key aspects and creating a coherent blueprint. It overcomes the scalability limits of planning, which is a problem not addressed by traditional planners [1]. This prevents performance degradation from information and constraints overload.
>
> → **Acceleration**: We introduce MAoP Distillation (Section 3.3) as an optional step to transfer the capability of this two-model system into a single, more efficient model (**one-step wide-horizon thinking**) for faster inference.
>
> **2. The Travel-Sim Benchmark (Travel with Your Plan):**
>
> → **Evaluation**: To evaluate these plans, we developed Travel-Sim. Existing benchmarks cannot evaluate the dynamic, sequential nature of travel where one event impacts the next (also illustrated as **consistency** in the paper). Travel-Sim solves this by having an LLM-powered traveler agent simulate the trip in a sandbox environment that uses real-world APIs.
>
>
> **W2:** The experimental section lacks sufficient description of the datasets, providing only macro-level descriptions.
>
> **A2:**
> Due to limited space, we provide the details of datasets in Appendix E, F and Figure 9. The Travel-Sim dataset (including the reward model training, RFT/RL training and evaluation datasets) comprises three key components: destination city data, preprocessed information, and traveler profiles.
>
> **1. Destination city data**
>
> We select cities from the list of China's Excellent Tourism Cities (a city title issued by the National Tourism Administration of China). For each dataset, we ensure selected cities have no overlap while comprehensively covering diverse geographical, cultural, and environmental characteristics.
>
> | Dataset| City List | City Count | Traveler Type Count |
> |--------------|-----------|------------|-------------------|
> | **Reward Model Training** | 桂林(Guilin), 青岛(Qingdao), 洛阳(Luoyang),  西双版纳(Xishuangbanna), 沈阳(Shenyang), 武汉(Wuhan), 南昌(Nanchang), 郑州(Zhengzhou), 长春(Changchun), 咸阳(Xianyang), 兰州(Lanzhou), 扬州(Yangzhou), 潮州(Chaozhou), 广州(Guangzhou) | 14 | 37 |
> | **RFT/RL Training** | 北京(Beijing), 重庆(Chongqing), 南京(Nanjing),  成都(Chengdu), 大同(Datong), 景德镇(Jingdezhen), 大连(Dalian), 杭州(Hangzhou), 北海(Beihai), 丽江(Lijiang), 昆明(Kunming), 太原(Taiyuan), 咸宁(Xianning), 深圳(Shenzhen), 香港(Hong Kong), 长沙(Changsha), 汕头(Shantou), 秦皇岛(Qinhuangdao), 恩施(Enshi), 天津(Tianjin) | 20 | 120 |
> | **Evaluation** | 拉萨(Lhasa), 三亚(Sanya), 上海(Shanghai), 哈尔滨(Harbin),  大理(Dali), 西安(Xi'an), 厦门(Xiamen)| 7 | 16 |
>
> **2. Preprocessed information**
>
> In Appendix C's preprocessing workflow, we gather real-world data through multiple channels, including:
>
> - Interactive agent simulations
>
> - Travel blog content analysis
>
> - Search engine data extraction
>
> - Map API integrations
>
> For instance, attraction information is systematically processed from both travel blogs and search engine results, while geographical data is collected through map APIs.
>
> **3. Traveler profiles**
>
> For traveler profiles, we begin by **manually** creating 30 distinct traveler archetypes based on real-world travel advice requests from blog posts. These initial profiles are then refined using Gemini-2.0-Pro-Exp-0205 to serve as a high-quality seed dataset. To expand the dataset, we employ Gemini-2.0-Pro-Exp-0205 for synthetic data generation. All synthesized profiles undergo **manual human review and filtering** to guarantee accuracy and diversity.
>
>
> **W3:** The paper lacks comparative experiments and fails to compare against other existing methods.
>
> **A3:** Our proposed method, Multiple Aspects of Planning (MAoP), is directly benchmarked against the dominant paradigm for complex travel planning in LLMs [1, 2, 3]: long-horizon thinking, as exemplified by CoT and CoT variants.
>
> Our experiments feature direct comparisons against baselines representing these existing methods. Specifically, we test against zero-shot long-horizon thinking approaches and their finetuned versions. The results consistently show that MAoP yields substantial performance improvements of 5% to 40% over these approaches across different models.
>
>
> **Q1:** What is the purpose of One-Step Wide-Horizon Thinking in Section 3.3? Experimental evidence is needed to support the rationale for adopting a one-step approach.
>
> **A4:** As we mention in Section 3.3, the implementation of MAoP requires two different models and a complicated planning process; therefore we propose one-step wide-horizon thinking to **accelerate and simplify** the MAoP process by distillation.
>
> Compared to traditional MAoP, **experimental results (Section 5.2.3)** show that MAoP distillation from advanced models to smaller models achieves substantial performance improvements besides the efficiency improvements.
>
>
> **Q2:** What is the detailed formulation of the reward function for RL/RFT?
>
> **A5:**
> In the RFT, we use the following reward function:
> $$
> R_{PER} = 2 * (PER_{normal} - 0.5), \text{where } PER_{normal} \in [0, 1].
> $$
> The $PER_{normal}$ score is the normalized PER score (0-100 normalized to 0-1), which is directly predicted by the reward model. We reject the sample that gets the average $R_{PER} < -0.2$ (PER score < 40) after sampling $N = 3$ times.
>
> In the RL, we add an additional format reward as follows:
> $$
> R_{format} = \begin{cases} 0, & \text{if the format is correct,} \\ -1, & \text{if the format is incorrect.} \end{cases}
> $$
>
> The overall reward for RL is:
> $$
> R_{overall} = R_{format} + R_{PER}.
> $$
>
>
> **Q3:** Could the preprocessing framework workflow be included in the main text to facilitate reader understanding?
>
> **A6:** We understand your concern that the illustration of the preprocessing framework helps to understand the method details. We exclude the preprocessing framework in the main text due to the following reasons:
>
> 1. **Not primary focus**: In this paper, we mainly want to discuss the exploration of $L^3$ planning problem and novel simulation-based evaluation. **The preprocessing workflow is an engineering issue. We avoid adding more content to confuse the reader.** If the readers are interested, they can read the full details in Appendix C.
>
> 2. **Space limitation**: Our preprocessing framework is complicated, with several LLM agents with different functions and hierarchical cluster-based algorithms. It is hard to clearly illustrate the details due to the limited space.
>
>
> **Q4:** What is the source of the synthetic data used for training the reward model in MAoP?
>
> **A7:** Please see W2A2 and for further details please refer to Reviewer J2GB W2A2.
>
>
> **Q5:** What is the purpose of the comparison in Section 5.2.4? How does this relate to your proposed Consistent Travel Simulation?
>
> **A8:** Because the events in the simulation happen consistently (not independently), the traveler will be influenced by the previous events and spontaneously adjusts the following schedule, leading to a different trajectory from the plan. **By comparing the trajectories between the simulation and the plan**, we can find out if the plan is feasible (the mechanism of FEA score). The analysis of spontaneous behaviours of in Section 5.2.4 also illustrates that our consistent simulation is close to the real-world travelling scenarios.
>
>
>
> **Q6:** Could you include an evaluation of the simulation itself? The current experiments do not demonstrate the reliability of the simulation component.
>
> **A9:** **Yes, we have included an evaluation of the simulation itself, providing detailed experimental setup in Appendix F.5.** As shown in the following table of PER score evaluation for "R1-Distill 7B (s.) + R1-Distill 7B (p.)", the evaluation of traveler agents has **high consistency with the human evaluation, with the agreement rate of 92% (PER-agg. ratio)**. We also identify that the PER-co (travel cost) metric exhibits a relatively higher level of inconsistency. We find that the traveler agents sometimes buy souvenirs but do not include them in the travel budget. In these cases, human evaluators give lower PER-co scores due to the budget overruns.
>
> |  | ex | it| ar | st | co | PER-agg. | Agree. rate|
> |------|---------------|----------|-------------------|--------------|---------|---------|---------|
> Simulation| 77.5 | 86.4 | 79.3 | 76.4 | 87.6 | 81.4 | - |
> Human | 69.2 | 83.7 | 73.4 | 74.2 | 74.5 | 75.0 | 92.1%|
>
> [1] Xie, Jian, et al. "Travelplanner: A benchmark for real-world planning with language agents." arXiv preprint arXiv:2402.01622 (2024).
>
> [2] Chen, Aili, et al. "Travelagent: An ai assistant for personalized travel planning." arXiv preprint arXiv:2409.08069 (2024).
>
> [3] Tang, Yihong, et al. "ItiNera: Integrating spatial optimization with large language models for open-domain urban itinerary planning." arXiv preprint arXiv:2402.07204 (2024).

---

> > ### Author Response · Authors · 2025-08-05
> > **Supplement for W3A3 about failing to compare against other existing methods.**
> >
> > To demonstrate the generalizability of our method, we conduct a new set of experiments on the ITINERA benchmark. To ensure a fair comparison, we benchmark our method, MAoP, against the best setting implemented in the ITINERA repository. We used the same LLM (Qwen 2.5-32B).
> >
> > | Method (Model) | RR ↑ (%) | AM ↓ (meters) | OL ↓ |
> > | :--- | :---: | :---: | :---: |
> > | **ITINERA** Qwen 2.5-32B | 28.4 | 101.3 | 0.37 |
> > | **Zero-shot wide-horizon thinkg w/ Artificial Guidance** Qwen 2.5-32B | 45.0 | 74.8  | 0.39 |
> > | **MAoP** Qwen 2.5-7B (s.) + Qwen 2.5-32B (p.) | 53.4 | 42.9 | 0.23 |
> >
> > Here are the metrics used in ITINERA:
> > - **Recall Rate (RR)**: the recall rate of POIs in the ground truth itinerary, which evaluates the accuracy of understanding user requests and recommending personalized POIs.
> > - **Average Margin (AM)**: the average difference per POI between the total distance of the generated itinerary and the shortest distance (via TSP).
> > - **Overlaps (OL)**: the number of self-intersection points in the generated itinerary. AM and OL measure spatial optimization for POI visit order, with lower values being better.
> >
> > As the results clearly indicate, our MAoP significantly outperforms the ITINERA baseline. Even zero-shot wide-horizon thinking with artificial guidance can achieve better performance than ITINERA.
> >
> > - The Recall Rate (RR) of MAoP is nearly doubled (53.4% vs. 28.4%), demonstrating MAoP's superior ability to understand complex user requests and identify relevant POIs.
> >
> > - The Average Margin (AM) of MAoP is reduced by more than half (42.9m vs. 101.3m), and Overlaps (OL) of MAoP are substantially lower (0.23 vs. 0.37), highlighting our method's advanced capabilities in spatial reasoning and route optimization.
> >
> >
> > This evaluation clearly validates the generalizability of MAoP, addressing the reviewer's concern. We will incorporate this benchmark evaluation and the corresponding analysis into the revised version of our paper to further strengthen our contribution.

---

### Official Review · Reviewer_J2GB · 2025-07-01

**Clarity:** 4
**Significance:** 3
**Originality:** 3
**Rating:** 4
**Confidence:** 3

**Summary:**

The paper addresses the L³ planning problem—long context, instruction, and output—via a novel two-stage framework (MAoP) combining high-level aspect routing with step-wise planning. It introduces Travel-Sim, a simulation-based evaluation environment with LLM agents, real-world constraints, and multi-dimensional feedback. A reward model trained on simulated data is used to further improve planning via RFT and RL. Experiments show consistent gains over strong baselines, highlighting the framework’s effectiveness in complex, multi-objective planning tasks.

**Questions:**

1. Have you measured the agreement between RM or PER scores and human ratings (or real travel diaries)? If not, how confident are you in their generalizability beyond your simulator?
2. Have you considered end-to-end reinforcement learning (RL) training of strategists, rather than relying solely on rejection sampling fine-tuning (RFT)?
3. If you were to include noisy but realistic actions (e.g., shopping, waiting, transit delays), would the simulation cost become intractable? Are you considering macro-actions or hierarchical planning to address this?

**Ethical Concerns:**

["NO or VERY MINOR ethics concerns only"]

**Final Justification:**

Thanks for authors’ rebuttal and most of my concerns have been effectively addressed. While I still hold a minor reservation regarding the practical deployment of the proposed method, I find the work to be well-motivated and solid. The rebuttal is comprehensive and solid, led me to raise the Quality and Clarity score. I maintain my borderline accept recommendation and look forward to seeing this work further developed.

**Limitations:**

Yes

**Paper Formatting Concerns:**

No major formatting issues observed.

**Quality:**

4

**Strengths And Weaknesses:**

Strengths
1. The paper clearly defines the L³ planning challenge—Long context, Long instruction, Long output—and grounds it in the concrete and impactful domain of travel itinerary planning. The formulation is broadly applicable to other multi-constraint tasks.
2. The MAoP pipeline introduces a two-stage (strategist → planner) design that supports controllable and wide-horizon reasoning.
3. The use of an event-driven sandbox (Travel-Sim) with LLM-based agents for realistic plan execution and feedback is novel and enables richer, behavior-grounded evaluation.

Weaknesses
1. The reward model is trained exclusively on Travel-Sim outputs and reused during optimization, risking self-reinforcing biases and reward hacking.
2. Evaluation is restricted to 7 Chinese cities and 16 traveler profiles, with no evidence of adaptation to unseen geographies, cultures, or constraints.
3. The fixed action space (transit, rest, dine, sightsee) omits common activities like shopping, queueing, or performances. TPSS may not correlate with actual user satisfaction, and certain travel patterns may be underrepresented.
4. The paper reports token-level simulation cost but omits end-to-end compute breakdowns across training and inference stages, hindering reproducibility assessment.
5. The approach is only validated within the Travel-Sim setup. There is no transfer experiment on existing travel planning datasets (e.g., TravelPlanner, ITINERA), which limits understanding of generalization to other task formats.

---

> ### Author Rebuttal · Authors · 2025-07-30
>
> We are very grateful for thorough review and highly insightful questions.
>
> **W1:** The reward model is trained exclusively on Travel-Sim outputs and reused during optimization, risking self-reinforcing biases and reward hacking.
>
> **A1:** We construct a separate training dataset for the reward model from the ones in RFT and RL. **To avoid the risk of reward hacking you are concerned with, we do not reuse the data (cities and traveler types) in the optimization for RL and RFT.** We exclusively select the 14 cities that are not included in the Travel-Sim training and evaluation dataset. We also additionally construct 37 traveler types specifically for this dataset.
> We provide the detailed data composition of these three datasets (reward model training, RFT/RL training, evaluation).
>
>
>
> | Dataset| City List | City Count | Traveler Type Count |
> |--------------|-----------|------------|-------------------|
> | **Reward Model Training** | 桂林(Guilin), 青岛(Qingdao), 洛阳(Luoyang),  西双版纳(Xishuangbanna), 沈阳(Shenyang), 武汉(Wuhan), 南昌(Nanchang), 郑州(Zhengzhou), 长春(Changchun), 咸阳(Xianyang), 兰州(Lanzhou), 扬州(Yangzhou), 潮州(Chaozhou), 广州(Guangzhou) | 14 | 37 |
> | **RFT/RL Training** | 北京(Beijing), 重庆(Chongqing), 南京(Nanjing),  成都(Chengdu), 大同(Datong), 景德镇(Jingdezhen), 大连(Dalian), 杭州(Hangzhou), 北海(Beihai), 丽江(Lijiang), 昆明(Kunming), 太原(Taiyuan), 咸宁(Xianning), 深圳(Shenzhen), 香港(Hong Kong), 长沙(Changsha), 汕头(Shantou), 秦皇岛(Qinhuangdao), 恩施(Enshi), 天津(Tianjin) | 20 | 120 |
> | **Evaluation** | 拉萨(Lhasa), 三亚(Sanya), 上海(Shanghai), 哈尔滨(Harbin),  大理(Dali), 西安(Xi'an), 厦门(Xiamen)| 7 | 16 |
>
>
> **W2:** Evaluation is restricted to 7 Chinese cities and 16 traveler profiles, with no evidence of adaptation to unseen geographies, cultures, or constraints.
>
> **A2:** Yes, we leverage the cities in China to verify the effectiveness of MAoP in travel planning. As China is a large country, we can select the cities with diverse cultures and geographies. The meticulous selection of cities and traveler types in the evaluation dataset is to ensure the generalization in unseen geographies, cultures, or constraints. To demonstrate the diversity of our city selection, we present the details of 7 cities from our evaluation dataset that exhibits maximum geographical, cultural, and environmental diversity:
>
> | City | Geo Direction | Location | Cultural Features | Climate Type | Terrain |
> |------|---------------|----------|-------------------|--------------|---------|
> | Lhasa | Southwest | Tibetan Plateau | Tibetan culture, Buddhist holy land | Plateau climate | Plateau |
> | Sanya | South | Southern Hainan Island | Tropical island culture | Tropical marine | Coastal plain |
> | Harbin | Northeast | Northeast Plain | Russian architecture, ice culture | Cold temperate continental | Plain |
> | Shanghai | East | Yangtze River Delta | International metropolis, modern commerce | Subtropical monsoon | Alluvial plain |
> | Xi'an | Northwest | Guanzhong Plain | Ancient capital, Silk Road origin | Warm temperate semi-humid | Loess plateau |
> | Dali | Southwest | Yunnan-Guizhou Plateau | Bai ethnic culture, plateau lakes | Plateau monsoon | Plateau basin |
> | Xiamen | Southeast | Southeast Coast | Minnan culture, garden on the sea | Subtropical marine | Island hills |
>
> **Although many cultures and geographies are still not covered, existing diversity in these cities is sufficient for evaluating the travel planning ability of LLMs. Therefore, expanding more cities is not our primary focus and will be considered in the future work.**
>
> **W3:** The fixed action space (transit, rest, dine, sightsee) omits common activities like shopping, queuing, or performances. TPSS may not correlate with actual user satisfaction, and certain travel patterns may be underrepresented.
>
> **A3:**
>
> **1. Fixed action space omits common activities like shopping, queuing, or performances?**
>
> The sightseeing action is a general concept that includes various activities happening in the POI. If the POI is a famous shopping spot, the sightseeing action can be shopping in the shopping mall. As our sightseeing event is simulated by referring to the real travel blog posts, the sightseeing agent can consider real-world events, e.g., the necessity of queueing (as there are many blog posts reminding to arrive early to get in line if the POI is very popular).
>
> **2. Correlation between TPSS and actual user satisfaction?**
>
> TPSS, also called the FEA (Feasibility) score in the paper, is specifically used to evaluate the generated plan's feasibility by comparing the trajectory in the plan with the trajectory in the simulation. **We specially design another metric, PER (Personalization) score, to evaluate the user satisfaction.** Therefore, we have these two metrics to evaluate different aspects of the generated plan.
>
>
> **W4:** The paper reports token-level simulation cost but omits end-to-end compute breakdowns across training and inference stages, hindering reproducibility assessment.
>
> **A4:** **We report detailed compute budgets and setups for training in Appendix E.1 and details for inference in Appnedix E.2.** We will release the code and open-source the dataset shortly, following an internal inspection of the dataset (as some of the travel blog posts within it were obtained via an internal corporate API). Anyone who adheres to the license in Appendix B may use our code and dataset for reproducibility.
>
> **W5:** The approach is only validated within the Travel-Sim setup. There is no transfer experiment on existing travel planning datasets (e.g., TravelPlanner, ITINERA), which limits understanding of generalization to other task formats.
>
> **A5:** In this paper, our primary focus is to explore the challenging $L^3$ planning problem that most LLMs fail to solve. Existing travel planning datasets (e.g., TravelPlanner [1], ITINERA [2]) provide cases containing **simple constraints and short contexts**, which are not suitable for $L^3$ planning problem evaluation. Therefore, we specially build Travel-Sim to provide the **hard cases**. Through the preprocessing framework in Appendix C, we collect heterogeneous user preferences by interactive agents and real-world information through search engines and map APIs, making the cases in Travel-Sim contain **complicated constraints and long contexts**.
>
> [1] Xie, Jian, et al. "Travelplanner: A benchmark for real-world planning with language agents." arXiv preprint arXiv:2402.01622 (2024).
>
> [2] Tang, Yihong, et al. "ItiNera: Integrating spatial optimization with large language models for open-domain urban itinerary planning." arXiv preprint arXiv:2402.07204 (2024).
>
>
> **Q1:** Have you measured the agreement between RM or PER scores and human ratings (or real travel diaries)? If not, how confident are you in their generalizability beyond your simulator?
>
> **A6:** **Yes, we have measured the agreement between PER scores and human ratings, providing detailed experimental setup in Appendix F.5.** As shown in the following table of PER score evaluation for "R1-Distill 7B (s.) + R1-Distill 7B (p.)", the evaluation of traveler agents has **high consistency with the human evaluation, with the agreement rate of 92% (PER-agg. ratio)**. We also identify that the PER-co (travel cost) metric exhibits a relatively higher level of inconsistency. We find that the traveler agents sometimes buy souvenirs but do not include them in the travel budget. In these cases, human evaluators give lower PER-co scores due to the budget overruns.
>
> |  | ex | it| ar | st | co | PER-agg. | Agree. rate|
> |------|---------------|----------|-------------------|--------------|---------|---------|---------|
> Simulation| 77.5 | 86.4 | 79.3 | 76.4 | 87.6 | 81.4 | - |
> Human | 69.2 | 83.7 | 73.4 | 74.2 | 74.5 | 75.0 | 92.1%|
>
> As the PER score has high consistency with human evaluation, **the RM that is based on the PER score is also aligned with human ratings.**
>
>
> **Q2:** Have you considered end-to-end reinforcement learning (RL) training of strategists, rather than relying solely on rejection sampling fine-tuning (RFT)?
>
> **A7:** **Yes, we have considered this limitation, as shown in the bottom footnote on the 4th page.** We do not further implement RL on the strategist model because the RL pipeline has to go through a frozen planning model to get rewards, making it hard to optimize.
>
> **Q3:** If you were to include noisy but realistic actions (e.g., shopping, waiting, transit delays), would the simulation cost become intractable? Are you considering macro-actions or hierarchical planning to address this?
>
> **A8:** Yes, we consider noisy but realistic actions in simulation without increasing the simulation cost by following implementations:
>
> 1. shopping & waiting: As we mention in W3A3, we leverage the realistic travel-blog post as the reference to simulate the real-world events, e.g., queueing, shopping.
>
> 2. transit delay: We collect realistic transportation data from the map APIs, e.g., Amap and Google Map. These map APIs have already considered the real-world traffic conditions, and we do not need to additionally simulate the transit delay.

---

> > ### Comment · Reviewer_J2GB · 2025-08-04
> > **Additional Feedback**
> >
> > Thank you for the detailed and thoughtful rebuttal. The authors addressed most of my concerns with clarity and strong supporting evidence. However, I would still like to see results on at least one existing benchmark (e.g., TravelPlanner or ITINERA). Since Travel-Sim appears to cover many of the evaluation dimensions present in these datasets, the proposed method should, in principle, perform reasonably well on them. Even a small-scale transfer evaluation would help demonstrate generalizability and position the contribution more clearly in relation to prior work.

---

> > > ### Author Response · Authors · 2025-08-05
> > > **Response to additional feedback**
> > >
> > > We thank the reviewer for this constructive suggestion. To demonstrate the generalizability of our method, we conduct a new set of experiments on the ITINERA benchmark. To ensure a fair comparison, we benchmark our method, MAoP, against the best setting implemented in the ITINERA repository. We used the same LLM (Qwen 2.5-32B).
> > >
> > > | Method (Model) | RR ↑ (%) | AM ↓ (meters) | OL ↓ |
> > > | :--- | :---: | :---: | :---: |
> > > | **ITINERA** Qwen 2.5-32B | 28.4 | 101.3 | 0.37 |
> > > | **Zero-shot wide-horizon thinkg w/ Artificial Guidance** Qwen 2.5-32B | 45.0 | 74.8  | 0.39 |
> > > | **MAoP** Qwen 2.5-7B (s.) + Qwen 2.5-32B (p.) | 53.4 | 42.9 | 0.23 |
> > >
> > > Here are the metrics used in ITINERA:
> > > - **Recall Rate (RR)**: the recall rate of POIs in the ground truth itinerary, which evaluates the accuracy of understanding user requests and recommending personalized POIs.
> > > - **Average Margin (AM)**: the average difference per POI between the total distance of the generated itinerary and the shortest distance.
> > > - **Overlaps (OL)**: the number of self-intersection points in the generated itinerary. AM and OL measure spatial optimization for POI visit order, with lower values being better.
> > >
> > > As the results clearly indicate, our MAoP significantly outperforms the ITINERA baseline. Even zero-shot wide-horizon thinking with artificial guidance can achieve better performance than ITINERA.
> > >
> > > - The Recall Rate (RR) of MAoP is nearly doubled (53.4% vs. 28.4%), demonstrating MAoP's superior ability to understand complex user requests and identify relevant POIs.
> > >
> > > - The Average Margin (AM) of MAoP is reduced by more than half (42.9m vs. 101.3m), and Overlaps (OL) of MAoP are substantially lower (0.23 vs. 0.37), highlighting our method's advanced capabilities in spatial reasoning and route optimization.
> > >
> > >
> > > This evaluation clearly validates the generalizability of MAoP, addressing the reviewer's concern. We will incorporate this benchmark evaluation and the corresponding analysis into the revised version of our paper to further strengthen our contribution.

---

> > > > ### Comment · Reviewer_J2GB · 2025-08-05
> > > > **Final Comment**
> > > >
> > > > Thanks for your rebuttal and no further clarifications are needed.

---

> > > > > ### Author Response · Authors · 2025-08-05
> > > > >
> > > > > Thanks for your kind support and for helping us improve the paper. We sincerely appreciate your valuable suggestions.

---

### Note · Authors · 2025-08-12

Dear Area Chair and Reviewers,

We are the authors of Paper 19455 and sincerely appreciate your time and effort during the review process. Your constructive feedback greatly strengthened our contribution to: **1)** solving the complex planning problem with parallel and conflicting information sources and constraints, and **2)** proposing a simulation-based evaluation method effective for complex planning tasks.

Below is a summary of the reviews and our responses:
- **Reviewer J2GB** praised our proposed method "**is broadly applicable to other multi-constraint tasks**" and highlighted that the evaluation method "**is novel and enables richer, behavior-grounded evaluation**", awarding our paper **the highest possible scores for quality and clarity**. To address the concerns about dataset composition, realistic adaptation, and implementation details, we clarify that we include the details in the appendix and expand evaluations to more benchmarks, which validate the generalizability.


- **Reviewer Jw3b** praised our planning method "**addresses broad-scope reasoning challenges**" and our simulation-based evaluation "**incorporates innovative metrics that provides solid support for experimental validation**". In response to concerns about paper organization and related issues, we explain the logic and organization of the paper and address the misunderstanding about one-step wide-horizon thinking and consistent evaluation.


- **Reviewer RxvN** appreciated our planning method "**represents a novel and useful direction for LLM-based planning systems**" and our evaluation method "**is a notable improvement over prior evaluation strategies**". To address concerns about human alignment and further application, we clarify with experiments in the appendix and provide more application examples.

- **Reviewer uXfj** recognized the "**strong real-world application and practical pipeline**" and "**contribution to the community**". To address the concerns about concept definition and open-source intention, we clarify the concept relationship to our methods and provide a detailed open-source plan.


We will incorporate these enhancements, along with your recommendations, into the final manuscript. Thank you and all reviewers for your invaluable feedback and expertise. **Additionally, due to the negative engagement from Reviewer Jw3b and uXfj, we regret that we could not fully resolve the reviewers' concerns by further clarification.**

Best regards,

Authors

---

### Decision · Program_Chairs · 2025-09-17

**Decision:**

Accept (spotlight)

**Comment:**

The submission proposes an LLM-assisted travel planning method and framework. The two main contributions are Multiple aspects of planning approach to travel planning, which involves training the MAoP model in three distinct steps (reward model, rejection sampling fine-tuning, RL planning) and  applying the model to planning where the query is a long question and context which are decomposed into multiple aspects for input to the model. A simulation environment was introduced, TravelSim, evaluated for being an adequate model of a human traveller, and used in experiments for assessment of travel planning via MAoP. The work relies on mature implementations and extensive empirical evaluations.

Initially, some reviewers had a number of concerns about the submissions, however, the authors addressed all reviewers' comments and doubts, provided results of additional experiments, and convinced the reviewers that the paper is worth a publication.